# Knowledge-driven design of solid-electrolyte interphases on lithium metal via multiscale modelling

Janika Wagner-Henke [1], Dacheng Kuai[2,3], Michail Gerasimov [1], Fridolin Röder [4], Perla B. Balbuena [2,3,5] & Ulrike Krewer [1]✉

Due to its high energy density, lithium metal is a promising electrode for future energy storage. However, its practical capacity, cyclability and safety heavily depend on controlling its reactivity in contact with liquid electrolytes, which leads to the formation of a solid electrolyte interphase (SEI). In particular, there is a lack of fundamental mechanistic understanding of how the electrolyte composition impacts the SEI formation and its governing processes. Here, we present an in-depth model-based analysis of the initial SEI formation on lithium metal in a carbonate-based electrolyte. Thereby we reach for significantly larger length and time scales than comparable molecular dynamic studies. Our multiscale kinetic Monte Carlo/continuum model shows a layered, mostly inorganic SEI consisting of LiF on top of $Li_2CO_3$ and Li after 1 μs. Its formation is traced back to a complex interplay of various electrolyte and salt decomposition processes. We further reveal that low local $Li^+$ concentrations result in a more mosaic-like, partly organic SEI and that a faster passivation of the lithium metal surface can be achieved by increasing the salt concentration. Based on this we suggest design strategies for SEI on lithium metal and make an important step towards knowledge-driven SEI engineering.

While the performance of conventional Li-ion batteries is pushed closer and closer to their theoretical limit, the demand for new cell chemistries for secondary batteries which can fulfill the ever-increasing requirements of the market rises. One promising candidate is the lithium metal battery, which combines a lithium metal negative electrode with a liquid or solid electrolyte and a positive electrode, e.g., based on sulfur[1]. Compared to Li-ion batteries with graphite electrodes, batteries with lithium metal electrodes promise multiple times higher energy densities and higher specific capacities[2]. However, a wide commercial distribution of this technology is still prevented by several remaining challenges regarding its safety and efficient cyclability. These are mainly caused by the high reactivity of lithium metal with liquid electrolytes and by uncontrolled dendrite growth

during cycling, which causes ongoing losses of active material and high kinetic losses[3,4].

One important strategy to solve this is the control and stabilization of the interfacial layer between the lithium metal electrode and the liquid electrolyte. As introduced by Peled[5], this Li-ion conductive layer is usually referred to as solid electrolyte interphase (SEI). In order to prevent uncontrolled SEI and dendrite growth, the SEI needs to show a wide range of properties[6]: Electrical insulation and simultaneously good ionic conductivity, low thickness to prevent cyclable Li and capacity loss, prevention of the direct contact of the lithium metal and the electrolyte, and good mechanical stability to accommodate the large volume changes of the metal electrode. The targeted design of such a beneficial initial SEI requires a detailed understanding of the

[1]Institute for Applied Materials — Electrochemical Technologies, Karlsruhe Institute of Technology, Karlsruhe 76131, Germany. [2]Department of Chemical Engineering, Texas A&M University, College Station, TX 77843, USA. [3]Department of Chemistry, Texas A&M University, College Station, TX 77843, USA. [4]Bavarian Center for Battery Technology (BayBatt), University of Bayreuth, Bayreuth 95448, Germany. [5]Department of Materials Science and Engineering, Texas A&M University, College Station, TX 77843, USA. ✉e-mail: ulrike.krewer@kit.edu

fundamental formation mechanisms and the resulting chemical SEI composition and morphology in dependence on the electrolyte composition.

A wide range of experimental methods is applied to characterize the SEI under a variety of experimental conditions[7]. While performance and lifetime data are accessible by cycling experiments, static methods, e.g., differential voltage analysis[8], and dynamic characterization, e.g., via impedance measurements or nonlinear-frequency response analysis[9,10], it is more intricate to investigate the SEI's chemical composition or morphology experimentally. For this purpose, measurement techniques such as time-of-flight secondary-ion mass spectrometry[11] or X-ray photoelectron spectroscopy depth profiling[11,12] have been applied since the 1990s. More recently, cryo-transmission electron microscopy was used to study the SEI formation on electrochemically deposited lithium at nm scale[13,14]. From these studies, in general, two possible SEI structures are discussed for graphite and lithium metal electrodes: The mosaic-like[15] and the layered nanostructures[12,16]. Often, a dense inorganic layer is observed close to the electrode interface, and a more porous organic layer is observed closer to the electrolyte. A combination with a mosaic-like nanostructure of different SEI components within multiple layers was also reported[7]. Although these experimental investigations allow descriptive insights into the chemical SEI composition and morphology, they struggle to identify and understand the underlying mechanisms. Moreover, the spontaneous SEI formation after the first contact of the lithium metal and a liquid electrolyte has hardly been studied so far[4]. This might be, because these processes occur within a ps to μs time scale and are therefore mostly inaccessible by experimental observations. A new recently published cutting method represents an important step towards better experimental characterization of the spontaneous SEI formation on lithium metal. This makes it possible to study the layer growth on pristine lithium metal and determine the influence of native passivation layers[17].

Theoretical calculations allow a complementary and detailed resolution of the governing processes inside a battery cell, enabling a targeted virtual optimization or tracking of the state of the battery and its components ranging from the atomistic to the macroscopic scale[18,19]. On a macroscopic scale, cell models like Single Particle models[20] or Newman-type pseudo-two-dimensional models[21] allow understanding the limiting processes during the cycling of a Li-ion battery. In order to incorporate the effect of the SEI on cell performance with respect to ionic conductivity, thickness, and cell resistance, there are some examples in the literature where these models were extended by a mechanistic model for the anodic surface layer[10,22,23]. These models also allow identifying SEI properties, and thus SEI and cell diagnosis, after parametrization to discharge and impedance spectra. Further macroscopic surface models containing a SEI mainly focus on the long-term growth regime and the related charge transport through the SEI. They can predict macroscopic properties such as SEI thickness or porosity depending on applied potential or storage time[24]. Other macroscopic models focus on SEI degradation and reformation at elevated temperatures or during the thermal runaway[25,26]. All of these continuum models have in common that they cannot predict the detailed chemical composition of the SEI or its nanomorphology and mostly neglect or lump the details of the underlying kinetic processes.

On the contrary, atomic-scale theoretical approaches like density functional theory (DFT), ab initio molecular dynamics (MD) and classical MD provide an understanding of the initial steps of SEI formation. DFT[27,28] and ab initio MD[27,29] allow the understanding of the chemical and electrochemical mechanisms and the prediction of the reaction energy profiles, but the simulated time is mostly below 100 ps. Modern machine-learning methods could accelerate the prediction of reaction energy profiles for large reaction networks[30,31]. Classical MD with adequate force fields are well suited to evaluate SEI reaction progress

over tens of nanoseconds time scales[32]. Moreover, different hybrid quantum-classical MD approaches have recently reported that yield SEI compositions and structures for a wide range of electrolytes[33] or allow the prediction of the effect of the electrochemical double layer on intercalation and deintercalation processes[34]. However, these approaches are too computationally expensive to model phenomena at extended length- and time scales as envisioned in this work[35].

As already pointed out by other authors[36,37], bridging these scales requires advanced mesoscale modeling approaches such as Kinetic Monte Carlo (kMC) models. This stochastic method overcomes the described limitations of atomistic modeling approaches while keeping detailed molecular information. In terms of battery research it was already applied to a number of different systems[37] such as classical lithium-ion batteries[38,39] and next-generation lithium-sulfur[40,41] or lithium-air[42,43] chemistries. On lithium metal, combinations of DFT and kMC were recently applied to evaluate the Li morphology evolution in porous electrodes[44] and the effect of external pressure on the void generation during stripping[45]. Moreover, two-dimensional kMC models were applied to investigate the SEI formation on graphite electrodes[38]. Based on this, 2+1-dimensional kMC models were developed, considering the SEI growth on a two-dimensional anode surface in the perpendicular direction. These were then coupled with Newman-type macroscopic battery cell models and applied to evaluate the interaction between molecular growth of the SEI in one direction and macroscopic processes within the cell[39,46,47]. More recently, Spotte-Smith et al.[31] presented a homogeneous spatially not resolved kMC model which incorporates extensive computational reaction networks that were in a previous step identified and parameterized by first principle calculations and machine-learning methods[48]. This model was used to study the competition between SEI-forming reactions at graphite electrodes during cycling. In our recent publication, we introduced the first full three-dimensional kMC model covering the initial SEI formation on lithium metal[49]. Thereby, we showed that our novel modeling approach can identify the SEI structure and chemical SEI composition with a full spatial resolution within the first 100 ns after immersion.

None of the previously mentioned models can yet model the SEI formation on lithium metal on a molecular resolution above the nanosecond time scale. Hence, the governing processes on the mesoscale of SEI formation on lithium metal remain poorly understood. However, such insights would be a big step towards a knowledge-driven design of the SEI on lithium metal electrodes. In this work, we will give a detailed and physically consistent analysis of the SEI formation and growth process on lithium metal on the μs-scale and evaluate how it can be influenced by macroscopic properties. Therefore, we will introduce several substantial extensions to our previously published kMC model[49], which drastically enhance its predictability, the accuracy of the results and extends the simulated time. This allows for sound physical conclusions on the SEI formation and growth at realistic times. As presented in Fig. 1, these extensions include a multiscale concept, in which we couple our kMC model with a continuum model after the example of ref. 50 for guaranteeing electroneutrality as well as the use of a consistent set of DFT-based energy parameters of the ethylene carbonate (EC) degradation. In our simulations, we explicitly track the spatial distribution of electrolyte components, reaction intermediates, SEI components, and lithium metal. From this, we analyze the time-resolved morphology and chemical composition of the SEI and conclude on the limiting processes of its formation. Our model approach also allows us to take unprecedented steps towards the model-based design of SEI structures by predicting the effect of (local) concentration changes of the liquid electrolyte consisting of EC, ethyl methyl carbonate (EMC), and $LiPF_6$. This allows us to understand the large variety in literature-reported structures, from layered to mosaic-like, and to suggest new strategies for the knowledge-driven design of the SEI.

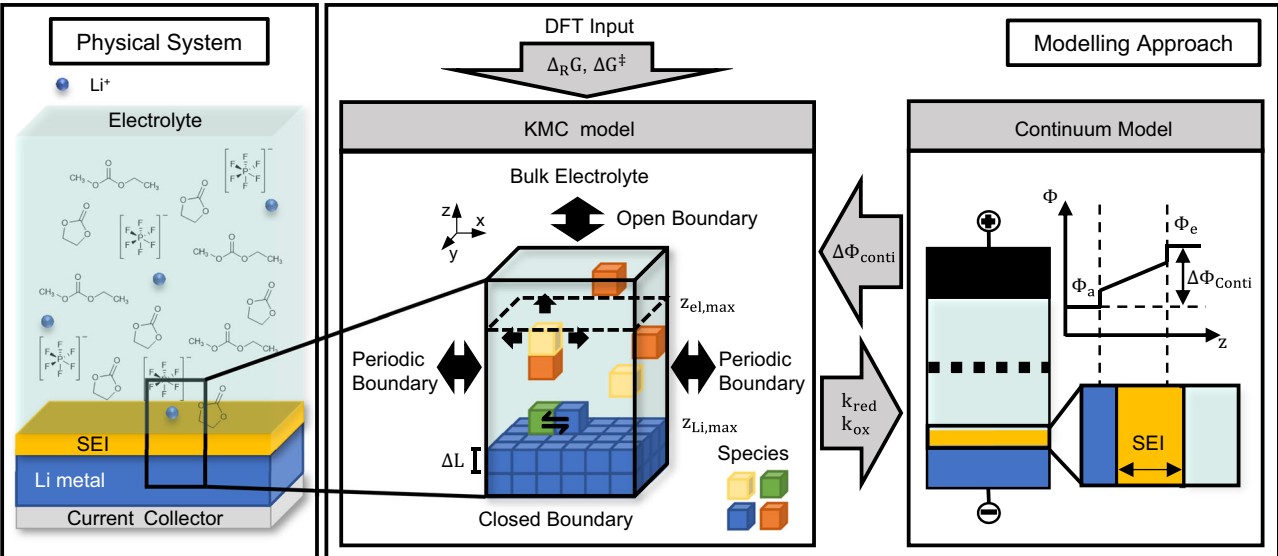

**Fig. 1 | Schematic modeling idea of SEI formation on lithium metal.** Left: Chemical system under investigation. Right: Multiscale kMC/continuum modeling approach with DFT input. DFT provides the Gibbs free energies $\Delta_R G$ and activation energies $\Delta G^{\ddagger}$ of the considered reactions for the kMC model. The kMC model passes the reaction constants $k_{red}$ and $k_{ox}$ to the continuum model and the continuum model returns the potential drop $\Delta\Phi_{conti}$ across the SEI.

## Results

### Parameter estimation and the effect of local Li$^+$ concentration

As a stochastic, mesoscale approach, the kMC model used in this study requires the input of a predefined reaction network as well as the related activation energies $\Delta G^{\ddagger}$ and free energies $\Delta_R G$ (cf. Fig. 1). The considered reaction network was determined based on literature information and DFT calculations. Thereby, we focus on electron transport reactions instead of adsorption reactions. This is due to the high reactivity of lithium metal which quickly leads to lithium oxidation and the loss of a distinct lithium metal surface on which electrolyte molecules could adsorb. A detailed derivation of the model and the reaction network can be found in "Methods".

Since the accuracy of the model and the predicted system dynamics highly depend on good-quality kinetic input data, we performed an extensive DFT study to identify consistent energy values for the local conditions at the lithium metal electrode. The resulting energies are summarized in Table 1. In the case of R2, which describes the ring-opening reaction of EC, no unique parameter set could be identified. It was recently shown in ref. 28 that the activation energy of this process is significantly impacted by lithium coordination. When coordinated with one Li$^+$ ion, the energy barrier was calculated to be 12.05 kcal/mol. As previously shown in literature[51], this barrier is mostly independent of the exact coordination ratio of EC and Li$^+$, and also applies if several EC-molecules are coordinated with one Li$^+$-ion. However, Kuai et al.[28] showed that the ring-opening barrier is significantly different for uncoordinated EC and becomes even negative. Similar declining energy surfaces are found in reactions R4 and R5. To transcribe this into the kinetic information, we assume the activation energies $\Delta G^{\ddagger}$ of the corresponding reactions to be 0 kcal/mol. Since the lithium coordination of EC and solvation are evolving dynamic processes in the electrolyte system, the energy barrier of the EC ring-opening process R2 is not clearly identifiable from the DFT calculations.

To estimate the effect of the ring-opening barrier on the resulting SEI structure and to choose a meaningful ring-opening energy for further investigations, we performed a parameter study with our multiscale SEI model. Details on the model, including the main processes of reaction, diffusion, and clustering, and the initial homogeneous distribution of liquid species in the electrolyte can be found in "Methods". For the parameter study, we vary the energy barrier $\Delta G^{\ddagger}$ of

the EC-ring-opening reaction R2 in the DFT-proposed parameter range between 0 and 12.05 kcal/mol (cf. Table 1). Thereby, each parameter set represents a different stochastic average of the availability of uncoordinated vs. Li$^+$-coordinated EC molecules close to the reaction site.

The results are shown in Fig. 2. In Fig. 2a, the final kMC box configurations after a simulated time of 1 µs are shown for different ring-opening energy barriers between 0 kcal/mol and 12.05 kcal/mol. In Fig. 2b, the corresponding composition of the resulting SEI is plotted. From this, we can see that the ring-opening energy has a strong effect on the resulting SEI composition and morphology. Low energy barriers lead to a SEI with an inorganic phase close to the lithium metal electrode and an organic phase above, closer to the electrolyte phase. The inorganic phase is composed of $Li_2CO_3$ and LiF, which are arranged in a mosaic-like manner, and the organic phase consists of $(CH_2OCO_2Li)_2$ (LiEDC). Moreover, $Li_2CO_3$ is strongly dispersed over the SEI, and its phase reaches inside the electrolyte. Relatively higher ring-opening barriers lead to a shift towards more LiF species in the inorganic phase. Moreover, both the organic phase and the dispersed $Li_2CO_3$ phase become less distinct until they entirely disappear when the ring-opening barrier reaches 12.05 kcal/mol.

Overall, these observations can be explained based on the reaction network shown in Supplementary Fig. 1. Since we tuned the kinetics of R2, the first reaction step of the EC degradation, it is reasonable that faster kinetics lead to more solvent-degradation-related products, which include both the inorganic $Li_2CO_3$ and the organic LiEDC. Vice versa, the increase of LiF quantities with slower EC degradation kinetics follows a complex interplay of diffusion and reaction limitations which is analyzed in detail in the next section. In brief, since fewer EC degradation products are formed, the passivation of the lithium metal is slower, which allows the reduction of salt over a longer period of time.

Another key finding is that increasing ring-opening energy barriers lead to a shift from a mosaic-like to a layered inorganic phase with $Li_2CO_3$ close to the lithium metal surface and LiF above. Interestingly, both the layered and the mosaic-like SEI structures on lithium metal were described in the literature and were experimentally observed[12,14–16,52]. Our new simulation results offer an explanatory approach for this variety of SEI structures reported in the literature for EC-based electrolytes: The kinetics of the EC degradation strongly

**Table 1 | Summary of implemented reaction processes and corresponding Gibbs free energies $\Delta_R G$ and activation energies $\Delta G^{\ddagger}$**

| N° | Reaction | $\Delta_R G$ (kcal/mol) | $\Delta G^{\ddagger}$ (kcal/mol) |
|---|---|---|---|
| R1 | $Li \rightleftharpoons Li^+ + e^-$ | $-70.22$[a] | $1.9$[a] |
| R2 | $EC \;\rightleftharpoons\; TS\text{-}R2 \;\rightleftharpoons\; LiEC$ | w/o Li⁺: $-30.43$[b] w/ Li⁺: $-39.17$[b] | w/o Li⁺: $0$[c] w/ Li⁺: $12.05$[b] |
| R3 | $2\,LiEC \;\rightleftharpoons\; TS\text{-}R3 \;\rightarrow\; (CH_2OCO_2Li)_2 + C_2H_4$ | $-56.5$[b] | $2.93$[b] |
| R4 | $LiEC + e^- \;\rightleftharpoons\; TS\text{-}R4 \;\rightarrow\; LiCO_3^- + C_2H_4$ | $-90.83$[b] | $0$[c] |
| R5 | $LiCO_3^- + Li^+ \;\rightleftharpoons\; Li_2CO_3$ | $-54.72$[b] | $0$[c] |
| R6 | $Li^+ + PF_6^- + e^- \;\rightleftharpoons\; LiF + PF_5^-$ | $-0.454$[d] | $3$[d] |
| R7 | $Li^+ + PF_5^- + e^- \;\rightleftharpoons\; LiF + PF_4^-$ | $-0.454$[d] | $3$[d] |
| R8 | $Li^+ + PF_4^- + e^- \;\rightleftharpoons\; LiF + PF_3^-$ | $-0.454$[d] | $3$[d] |

Energy values were (a) assumed, (b) calculated by DFT, (c) manually set to 0 kcal/mol since the DFT calculations suggested negative values or (d) adapted from ref. 49. TS represents the transition states. Reaction energies for R2 are given for two cases: with (w/) and without (w/o) lithium coordination of EC.

differ with the local lithium coordination and the local solvation environment. Further, the lithium coordination of EC depends on the local availability of Li⁺ and hence on the local Li⁺ concentration. Therefore, a high local Li⁺ concentration close to the electrode surface stabilizes many EC molecules and consequently leads to less EC degradation products such as LiEDC or $Li_2CO_3$ and a layered inorganic initial SEI layer. In contrast, a low Li⁺ concentration increases the availability of uncoordinated EC molecules and thus—due to the negligible barrier for uncoordinated EC—facilitates EC degradation. The resulting SEI contains more EC degradation products with a more mosaic-like morphology.

These local properties could vastly differ depending on the working conditions. Therefore, SEI formation on pure lithium metal under OCV conditions could be very different from SEI formation during lithium plating and the cycling of full cells. We expect that these observations are, in principle, transferable from lithium metal to lithium-ion batteries. However, it should be considered that the local Li⁺ concentrations at the interface could be very different in both systems. The lithium metal electrode provides a large source of Li⁺ via oxidation, while the intercalated ions in graphite electrodes show a comparatively lower availability under open-circuit voltage conditions due to a higher Open-Circuit Potential[53] and charging-induced passivation layers[54]. This could be a root cause for different SEI morphologies in both systems. Overall, the control of the local Li⁺ concentration close to the interface is an interesting new strategy in rational tuning the SEI on lithium metal but also on intercalation electrodes for Li-ion batteries.

In order to better understand which parameter set represents best the conditions at the initial interface between lithium metal and liquid electrolyte, we compare our simulation results in the following with the data of the recent MD study of Ospina-Acevedo et al.[32]. These authors applied reactive MD simulations with ReaxFF to model the first 20 ns of SEI formation in different electrolytes—among others, the EC + $LiPF_6$ electrolyte, which we consider in our study. Since MD and kMC are very different simulation paradigms, which were applied to the same chemical system with identical size, time, and temperature, the comparison is a good approach for validating and benchmarking our modeling approach. Moreover, comparison shows the low computational costs of our approach: according to the authors of the comparative MD study[32], their calculations ran for a couple of weeks on high-performance computer clusters in order to reach 20 ns. In contrast, our kMC/continuum model only took 29.2 min on a personnel computer with an i7-8700 CPU and 16 GB RAM to reach the same time on a 32 times larger length scale.

The comparison of the resulting MD box and the kMC/continuum simulation with a ring-opening energy of 12.05 kcal/mol after 20 ns is shown in Supplementary Fig. 2. Despite the difference in length and time scale of the MD and the kMC/continuum simulations, a qualitative comparison can be made by using a representative section of the lithium metal surface in the kMC box after a simulated time of 20 ns.

First of all, the MD simulation suggests a layered inorganic SEI. Thereby, below the initial interface carbonate, ethene, $LiO_x$, $PF_3$, and $CO_2$ can be found. Above the initial interface, a LiF layer can be observed. Moreover, no organic species were formed within the simulated time period. Please note that in the study of Ospina-Acevedo et al.[32], the salt was initially placed close to the lithium metal surface while it was evenly distributed in our kMC/continuum study. In any case, both the layered structure and the absence of organic SEI components are qualitatively similar to our kMC/continuum simulation results for high EC ring-opening energies of 10 or 12.05 kcal/mol.

For a more quantitative comparison, we identified the number of released ethene molecules as a good indicator for the EC degradation kinetics. Since it can only be produced as a side product of the EC degradation, it is directly related to the number of reduced EC molecules and can be quantified by the tracked number of the ethene-

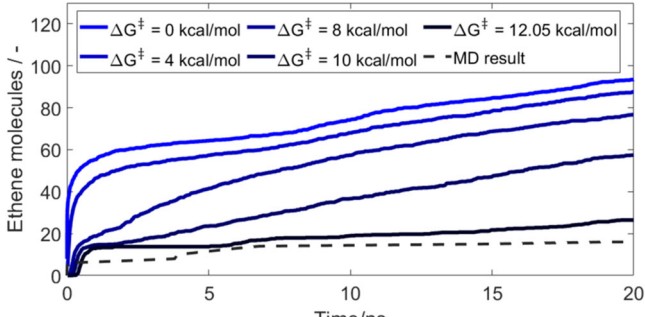

**Fig. 2 | SEI composition and structure 1 μs after the initial contact of EC + 1.2 M LiPF₆ and lithium metal for varying EC ring-opening energies ΔG‡. a** Resulting species distribution in kMC box. **b** Number of SEI molecules over the height. The dashed lines represent the initial lithium metal surface (left) and the maximum electron transport distance (right). Source data are provided as a Source Data file.

forming reactions R3 and R4. In order to ensure comparability between both simulation approaches, the released gas in the kMC/continuum simulations was scaled by the scaling factor $\nu = \frac{n_{EC,MD}^0}{n_{EC,kMC}^0}$, which relates the number of initial EC molecules in the MD $n_{EC,MD}^0$ simulation box to the number of initial EC molecules within a distance of 2 nm above the lithium metal in the kMC simulation box $n_{EC,kMC}^0$. In Fig. 3, the result is plotted over the simulated time. Thereby, only the ethene production for the parameter set with an EC ring-opening energy of 12.05 kcal/mol, which corresponds to the ring-opening energy with Li⁺ coordination, is in the same order of magnitude as the ethene production in the MD simulation. However, the ethene release for this parameter set is still slightly higher than observed from the MD calculations. This effect could have multiple reasons. First of all, all salt ions were placed close to the lithium metal surface in the MD simulation. This could lead to an overestimated LiF production, which quickly passivates the surface and hinders fresh EC molecules from reaching the lithium metal surface. Moreover, the kMC/continuum simulation has an open upper boundary connected with a bulk electrolyte phase, while in the MD simulation box all considered molecules are placed in the initial box. This means that in the case of the kMC/continuum simulation, a higher amount of fresh EC is available to be transported towards the lithium metal surface.

**Fig. 3 | Comparison of the number of released ethene molecules over time for varying EC ring-opening energies ΔG‡ with MD simulation performed by Ospina-Acevedo et al.[32].** The number of ethene molecules for the multiscale kMC/continuum simulation is calculated from the occurrence of the gas-forming reactions R3 and R4 and is scaled to the amount of EC molecules in the initial MD simulation box. Source data are provided as a Source Data file.

Overall, the comparison with the MD simulation clearly indicates that the kMC/continuum simulation using the highest EC ring-opening barrier of 12.05 kcal/mol, which corresponds to a strong influence from

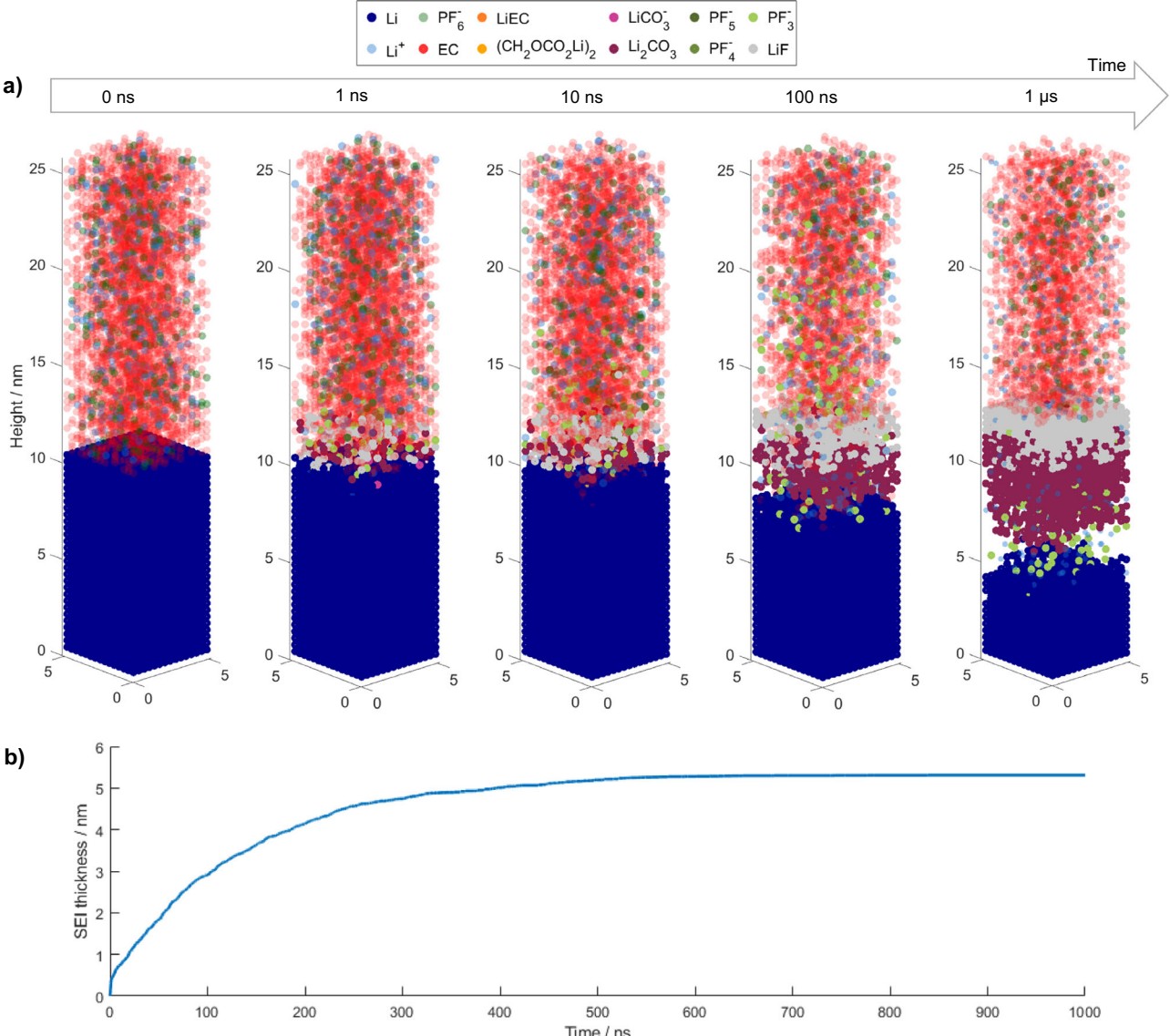

**Fig. 4 | Temporal evolution of SEI formation on lithium metal in EC + 1.2 M LiPF_6 electrolyte. a** Resulting species distribution in kMC box at selected times and **b** development of SEI thickness over time. Energy values comprise the case for EC reduction in the presence of Li$^+$. Source data are provided as a Source Data file.

local Li$^+$, best reproduces the local conditions at the lithium metal surface. Since the lithium metal electrode acts as a source of Li$^+$ ions, which are the cause for the EC ring stabilization, this result is reasonable for the investigated cell chemistry. Therefore, for the following detailed SEI analysis, including the influence of salt and solvent concentration on the SEI, we apply an EC ring-opening barrier of 12.05 kcal/mol.

**Temporal analysis of SEI formation**
The following analysis of the temporal evolution of the SEI formation allows an in-depth understanding of the limiting processes and process interactions that cause the final SEI composition and thickness. The general trends of SEI formation over time are presented in Fig. 4, which shows several snapshots of the kMC box along with the development of the average SEI thickness over time. Thereby, the SEI thickness shows an asymptotic growth to 5.3 nm, where 50% and 90% of the height is reached after ~100 and 200 ns, respectively.

In all, 1 ns after the initial contact of lithium metal with the liquid electrolyte, the first intermediate products and SEI species form within 1 nm from the initial surface. This shows the high reactivity of pure lithium metal, which was already described by He et al.[4]. As reduced lithium with a very low open-circuit potential is present from time $t = 0$,

the SEI formation starts immediately after contact of the electrolyte with the electrode material and does not require any external current. After 10 ns, some more SEI species were formed, and the initiation of the layering of the inorganic phase, with a Li$_2$CO$_3$ layer below a LiF layer, can be observed. The two layers of the inorganic phase can be clearly distinguished after 100 ns. It is interesting to note that LiF mainly forms above the initial solid-liquid interface at the height of ~10 nm. In contrast, Li$_2$CO$_3$ grows into the former lithium metal phase, and is mainly present below the initial interface. In comparison with the SEI after 1 μs, both observed phases are still very porous, and solvent molecules are still present close to the lithium metal surface. Eventually, after 1 μs all EC molecules below the newly formed SEI layer are consumed. Moreover, the LiF and the Li$_2$CO$_3$ layer became less porous and thicker. Thereby, the LiF layer mainly grows towards the electrolyte, while the Li$_2$CO$_3$ layer further grows into the lithium metal phase below the initial interface. As already described above, even after 1 μs, we cannot observe the formation of organic species for this parameter set. This suggests that the experimentally observed organic SEI species[4] form only on larger time scales and require an initial inorganic passivation of the lithium metal surface. The same SEI formation and chemical SEI composition were observed when smaller

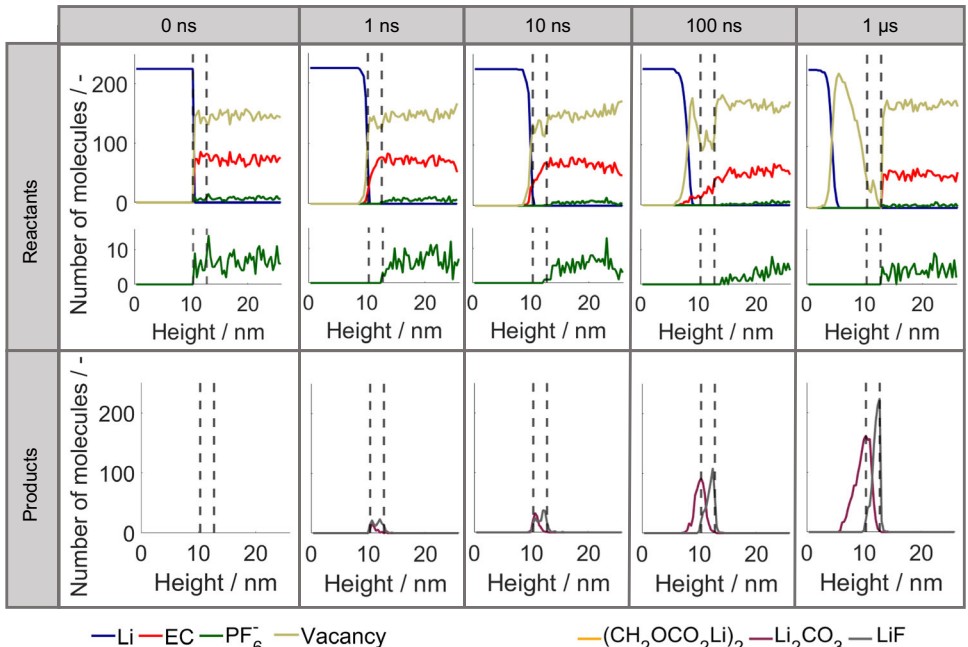

**Fig. 5 | Temporal evolution of the species distribution over the height of the kMC box.** The first row shows the reactant distribution, and second row shows the SEI products distribution at selected time points. The dashed lines represent the initial lithium metal surface (left) and the maximum electron transport distance (right). Energy values comprise the case for EC reduction in the presence of Li[+]. Source data are provided as a Source Data file.

inhomogeneities were introduced to the lithium metal surface, as shown in Supplementary Fig. 3. This shows the low sensitivity of the principle SEI layer on structural inhomogeneities. Since the SEI is so thin, the layer structure can thus be expected also for Li interfaces with surface roughness. Detailed experimental studies on the initial SEI formation on lithium metal and its resulting composition, especially with a sub-μm resolution, are scarce in literature[4,55]. To the best of our knowledge, there is no experimental study, yet, which was able to reveal the here observed layering of the inorganic SEI. Future advancements in experimental methods may allow to reach a similar resolution and thus provide experimental validation.

In addition, it is interesting to note that, in comparison to the first nanosecond, the SEI growth within the subsequent 9 ns slowed down significantly. This can also be seen in the thickness evolution in Fig. 4b. After a sharp increase within the first ns, the SEI growth continuously slows down until it reaches a maximum thickness of 5.3 nm after ~700 ns. This deceleration is caused by the increasing surface passivation constituted by the previously formed SEI species. After 700 ns, the surface becomes completely passivated to further electron tunneling, which is considered the only electron transport process in our model. Indeed, this does not mean that no further growth of the SEI is possible. Additional electron transport processes such as solvent diffusion, electron conduction through the SEI or Li-interstitial diffusion could lead to ongoing SEI growth on significantly larger time scales[27,56]. These are not considered in our model since we are not expecting a substantial effect during the first μs of SEI formation. Moreover, the cycling current will also lead to changes in and further growth of the SEI. Last but not least, the developing electrical double layer could affect the SEI formation. Its effect on our modeling results is analyzed in the Supplementary Information in section 7.

The obtained Li$_2$CO$_3$-LiF-layered SEI partly differ from the observations made in our prior work[49]. Despite a reported layered SEI, the order of the inorganic species was different, and a very fast formation of organic species within the first 100 ns after the initial contact of lithium metal and the liquid electrolyte was observed, which cannot be confirmed here. We mainly attribute these differences to the overestimation of the reaction rate of the EC degradation in ref. 49. The

activation barrier of the EC ring-opening reduction was estimated at 2.7 kcal/mol, while this work found a barrier of 12.05 kcal/mol by considering the stabilization effect of Li[+] ions on the EC-ring. This is why the previously predicted SEI is more similar to our results with lower EC ring-opening energies of 0 or 4 kcal/mol, which can be found in Fig. 2a, b (left). Another crucial difference is that the more detailed energetic calculations in this work have found that the energy barriers of the possible subsequent degradation reactions are the reverse of what was assumed by the previous study. While in this work, the formation of inorganic carbonates was found to be more likely in an electron-rich environment due to lower energy barriers of the carbonate-forming reaction R4 compared to the organic-forming reaction R3, the formation of organic LiEDC was energetically preferred in the former study of Gerasimov et al.[49]. Overall, the inclusion of electroneutrality, including multiscale coupling, the reparameterization and further model extensions done in this work lead to a significant improvement in the accuracy and predictability of our modeling approach.

In order to understand the underlying processes governing SEI formation, a detailed analysis of the temporal changes in the distribution of reactants and SEI products is presented in Fig. 5. Since the salt concentration is comparatively low, for better visualization, it is plotted in a separate figure with a zoomed-in y axis. From these figures, it is well visible that the salt concentration within the electron transport zone (between the dashed lines) quickly drops to 0 within 1 ns. This results from the rapid salt degradation kinetic with an energy barrier of only 3 kcal/mol (cf. Table 1). Since this process is approximately one order of magnitude faster than the salt diffusion, all available salt ions are reduced at their current position. Hence, the degradation product LiF is evenly distributed throughout the electron transport layer at this time point of the simulation. Subsequent diffusion of PF$_6^-$ from the neighboring electrolyte layers to the electrochemically active reaction zone leads to a concentration gradient into the electrolyte zone, as visible for 10 and 100 ns. This indicates that all salt which reaches the electrochemically active zone is immediately consumed. The development of the LiF concentration over time confirms this observation since it develops an increasing concentration peak at the upper end of the electron transport layer. Overall, this shows that the transport rate

of the reactant $PF_6^-$ is slower than its decomposition. Hence, we conclude that LiF production is a diffusion-limited process.

At the end of the simulation, after 1 μs, the concentration gradient above the electron transport zone mostly vanished. This matches the observation made in Fig. 4b that the interphase is thicker than the electron tunneling zone, which blocks the electron tunneling to a negligible probability. Hence, no new salt above the formed SEI is consumed after 1 μs. The consumption of EC, which is approximately 11 times more numerous in the electrolyte solution, is much slower in comparison. This can be especially observed in the EC concentration profiles. Even after 100 ns, solvent is still present within the electron transport layer. Moreover, it should be noted that EC molecules even exist below the initial interface after 10 and 100 ns. This relatively high stability of EC can be traced back to the energy barrier of the first EC reduction step. This barrier was determined to be 12.05 kcal/mol, which is high compared to the energy barrier of only 3 kcal/mol for the salt decomposition (cf. Table 1). In addition, in contrast with the salt concentration profiles, no EC-concentration gradient develops above the electron transport layer. From this, we can conclude that the first step of electrolyte degradation is reaction-limited. With regard to the presented mesoscale calculations, this means that the EC molecules have enough time to approach the lithium metal surface without being immediately reduced. This further allows the formation of the $Li_2CO_3$ peak closer to the lithium metal surface and below the LiF layer.

The reason why $Li_2CO_3$ is preferably produced over LiEDC can be explained by comparing the kinetics of both EC degradation pathways in R2-R4. As previously pointed out by Yu et al.[57], the second EC reduction step R4 which reduces LiEC to $LiCO_3^-$ is faster than the first EC reduction step R2, which reduces EC to the ring-opened LiEC. This is also reflected in our kinetic parameters (cf. Table 1), in which the activation barrier of R2 is 12.05 kcal/mol, and the barrier for R4 is 0 kcal/mol. Hence, as long as electrons are readily available, which they are close to the Li surface, newly formed LiEC can quickly undergo a subsequent reduction reaction via reaction R4, leading to low LiEC concentrations and significant $LiCO_3^-$ production. In contrast, two LiEC at neighboring sites are needed to produce LiEDC (cf. R3). As the LiEC concentration is low, this makes LiEDC production unlikely close to the surface. Hence, the production of LiEDC could only occur in an electron-deficient environment, which prevents the second reduction step. One possibility to obtain this environment is a first passivation layer at the surface which significantly slows down the electron leakage from the lithium metal electrode. This explains why the organic SEI species are often observed at the top of the SEI: They require a first passivation layer to be formed. Therefore, we would expect a formation of organic species such as LiEDC mostly on larger time scales.

Meanwhile, Fig. 5 also answers the question of why the surface is fully passivated against electron tunneling at the end of our simulation. From the reactant distribution after 1 μs, we can see that the number of vacancies drops to zero at the upper electron transport limit, i.e., at the interface to the electrolyte. This means that the surface is fully occupied with SEI species. Hence, solvent and salt molecules in the electrolyte phase cannot pass the outer SEI surface and thus are not close enough to the lithium metal to be reduced by electron tunneling. Further, at 1 μs, all reactants inside the electron transport range are consumed. Therefore, no further SEI formation is possible. Transferring this insight to the actual physical system of 1.2 M $LiPF_6$ in EC/EMC, we would expect that this initial passivation significantly slows down further SEI growth, which could continue only by other, slower transport paths, such as interstitial Li diffusion or electron diffusion[27,56].

## Influence of salt and solvent concentration

Having revealed the SEI composition and related governing processes of the SEI formation on lithium metal in EC with 1.2 M $LiPF_6$, we will now use the model to show how to modify the resulting SEI by tuning macroscopic properties.

An intuitive approach is to vary the concentration of the conductive salt in the electrolyte. Therefore, we studied the SEI formation for four different $LiPF_6$ concentrations from 1 M to 4 M. In doing so, we applied the same set of kinetic parameters for all salt concentrations and hence neglected possible salt concentration effects on the reaction kinetics. The results are presented in Fig. 6. The corresponding kMC-boxes after 1μs can be found in Supplementary Fig. 4.

At first sight, it seems surprising that the change of the salt concentration has only a minor effect on the distribution and quantities of LiF, while the amount of the EC degradation product $Li_2CO_3$ decreases. However, considering the previously analyzed governing processes of the SEI formation, this observation can be explained as follows: We learned that the salt degradation process which produces LiF is diffusion-limited, and that $PF_6^-$ anions which reach the electron transport layer are almost immediately consumed. An increased salt concentration in the electrolyte automatically results in a higher initial salt concentration in the electron transport zone (cf. Supplementary Table 2). These salt species are almost immediately reduced and thus increase the LiF concentration in the electron transport layer. Moreover, the transport of additional salt species from the electrolyte solution towards the lithium metal surface is more frequent in systems with higher salt concentrations. Overall, this leads to a faster production of LiF and hence to a quicker passivation of the lithium metal surface. As a consequence, fewer solvent molecules can approach the electron transport layer before the complete passivation of the surface. This can be confirmed by the ratio of initially present to overall consumed EC, which is summarized in Supplementary Table S2 for all parameter sets. This ratio increases from 0.116 for 1 M $LiPF_6$ to 0.193 for 4 M $LiPF_6$ and hence indicates that fewer EC molecules diffuse to the electron transport zone for high salt concentration. The decreased availability of solvent close to the lithium metal surface results in the observed reduction of the EC degradation product $Li_2CO_3$. Furthermore, it leads to increased porosity of the inner SEI, which can be seen in Fig. 6b.

The described faster passivation of the SEI surface against transport processes can also be observed in Fig. 6c, which compares the SEI thickness evolution for the investigated salt concentrations. Higher salt concentrations lead to steeper growth of the initial SEI layer. At the same time, the final mean SEI thickness decreases with the salt concentration. From the final kMC boxes, which are shown in Supplementary Fig. 4, we can additionally conclude that less lithium metal is consumed for the fast formation of this first passivation layer.

Overall, by modifying the salt concentration in the electrolyte, the ratio of $Li_2CO_3$ and LiF in the inorganic layer of the SEI, the thickness of the inorganic layer, and the loss of active material can be purposefully tuned. From this, we conclude that higher concentrations of the conductive salt lead to a faster passivating SEI with a higher LiF content. This is well in line with literature, which suggests that a high LiF concentration in the SEI can enhance the self-protection of the SEI and mechanical stability[3,58].

EC is usually mixed with further, more chemically stable solvents such as EMC, DMC or PC[59]. To see the effect of adding such a more stable solvent compound to the electrolyte, we can use the here presented model. For our study, we exemplarily use EMC as the second electrolyte solvent, which dilutes the EC concentration. As discussed in more detail in the methods section, we do not expect a significant number of EMC degradation products to be formed within the first microsecond and therefore did not consider any related reduction reaction. We further assume that the second solvent does not have a significant impact on the considered reaction and transport processes and does not chemically interact with the salt or co-solvent in the investigated period of time. Under these assumptions, the following results, which are presented in Fig. 7, are valid for each electrolyte solvent with a higher or similar chemical stability as EMC on lithium metal. The corresponding kMC boxes after 1 μs can be found in Supplementary Fig. 5.

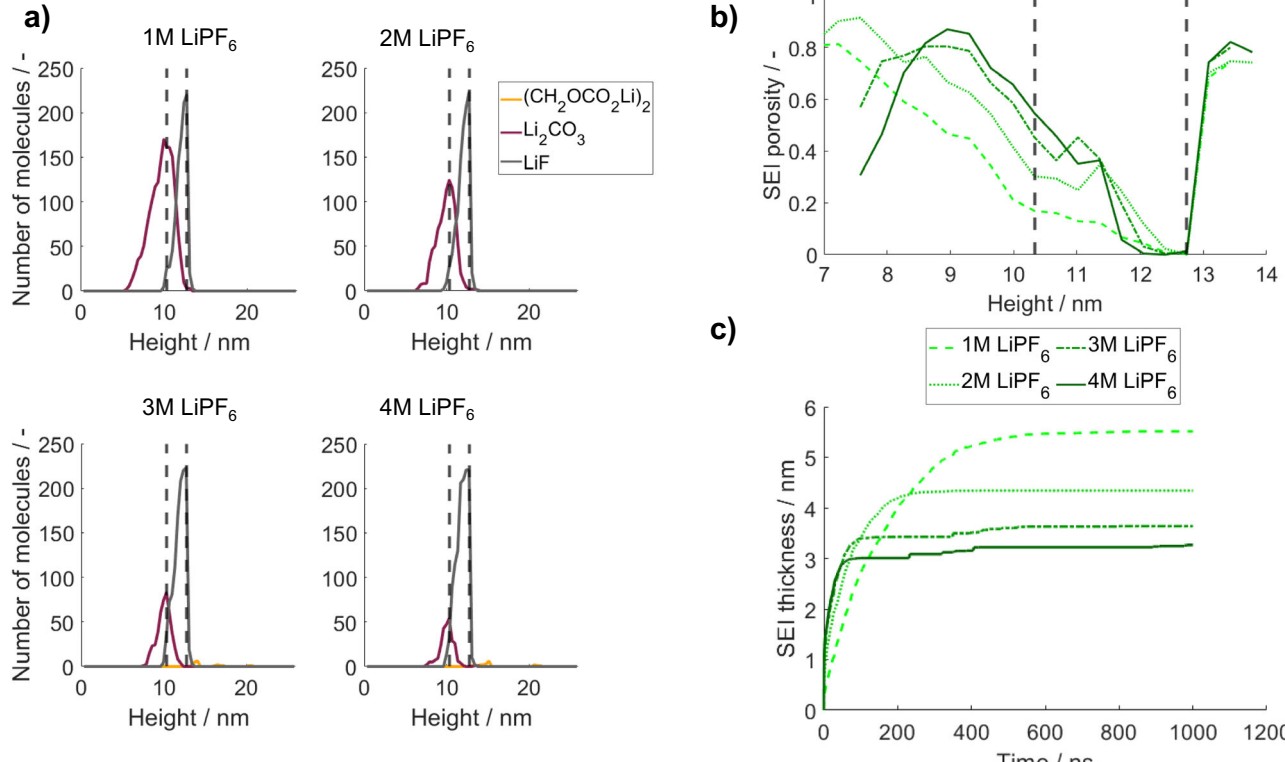

**Fig. 6 | Variation of the salt concentration between 1 M and 4 M LiPF$_6$ in 100 wt % EC as solvent. a** Number of SEI molecules after a simulated contact time of Li with the electrolyte of 1 μs over height. The dashed lines represent the initial lithium metal surface (left) and the maximum electron transport distance (right). **b** SEI porosity after 1 μs over height for the investigated salt concentrations. **c** SEI thickness evolution for the investigated salt concentrations within 1 μs. Source data are provided as a Source Data file.

The simulation results show the following effect of a decreased EC concentration: The LiF distribution in the formed SEI does not change significantly while the amount of formed Li$_2$CO$_3$ decreases. Overall, the inorganic SEI keeps a layered structure. Moreover, the thickness of the SEI decreases with decreasing EC concentration. These observations can be mainly attributed to the lower amount of EC molecules initially present close to the lithium metal surface at decreased EC concentrations (cf. Supplementary Table S2). In addition, the transport of fresh reactants towards the electron transport layer is decelerated since fewer EC molecules are available in the electrolyte solution. As a consequence, less Li$_2$CO$_3$ can be formed before the surface becomes fully passivated by LiF. Interestingly, the ratio of initially present vs overall consumed EC molecules (cf. Supplementary Table S2) decreases with reduced EC concentration. This means that a greater proportion of the consumed EC is transported towards the electron transport zone with lower EC concentrations. In addition, from Fig. 7b, we can see that the porosity of the inner SEI increases with decreased EC concentrations.

Although these observations seem similar to the effect of the salt variation, it should be noted that in contrast to the sensitivity to salt concentration, the speed of the passivation cannot be tuned by adjustment of the EC concentration. This important difference can also be observed in Fig. 7c, which shows that the SEI thickness evolution is consistent for all solvent concentrations within the first nanoseconds. Afterward, it becomes first limited for low EC concentrations since fewer fresh EC molecules are available in the electron transport zone.

This difference between the effect of the salt and solvent concentration can have some important implications in real systems. In general, real experimental systems are usually more complex than the ideal assumptions in our model. Neither the lithium metal electrode nor the electrolyte components are perfectly clean. Therefore, impurities within the electrolyte and lithium metal or the often-reported initial surface layer on lithium[60] may cause additional effects and modify the outcome. Furthermore, EMC may also degrade, though slower, leading to additional degradation products. Hence, fast passivation of the lithium metal surface is preferable since it could prevent many of these undesirable processes and could decrease side products from impurities. Overall, a faster passivation would lead to a more controllable SEI. Therefore, we suggest to preferably tune the inorganic initial SEI by a variation of the salt concentration. Thereby, also the known downsides of high salt concentrations, such as slower ion transport, high viscosity and cost need to be considered in order to identify the optimal electrolyte composition.

## Discussion

A novel kMC/continuum multiscale modeling approach was used to obtain detailed insight into the procedure of SEI formation on a 50 times larger time scale and a 32 times larger length scale than comparable ReaxFF MD simulations[32]. Thereby, the presented approach is able to track the time-evolution of SEI formation on a molecular resolution and to reveal details of the SEI composition and morphology and on the underlying formation mechanisms, which are presently inaccessible by experiments[4,55].

During the initial SEI formation, the transport was identified to be the limiting process for the degradation of the conductive salt LiPF$_6$, whereas the decomposition of the electrolyte solvent EC is limited by its reaction kinetics. In this context, the carbonate electrolyte solvent can maintain its chemical integrity for a sub-microsecond time scale when approaching the lithium metal surface. Overall, we revealed a complex interplay of electrolyte diffusion, electrolyte degradation and consumption of the lithium metal electrode, which results in a layered inorganic SEI on the lithium metal surface. Thereby, a Li$_2$CO$_3$ layer was observed close to the lithium metal surface and a LiF layer above.

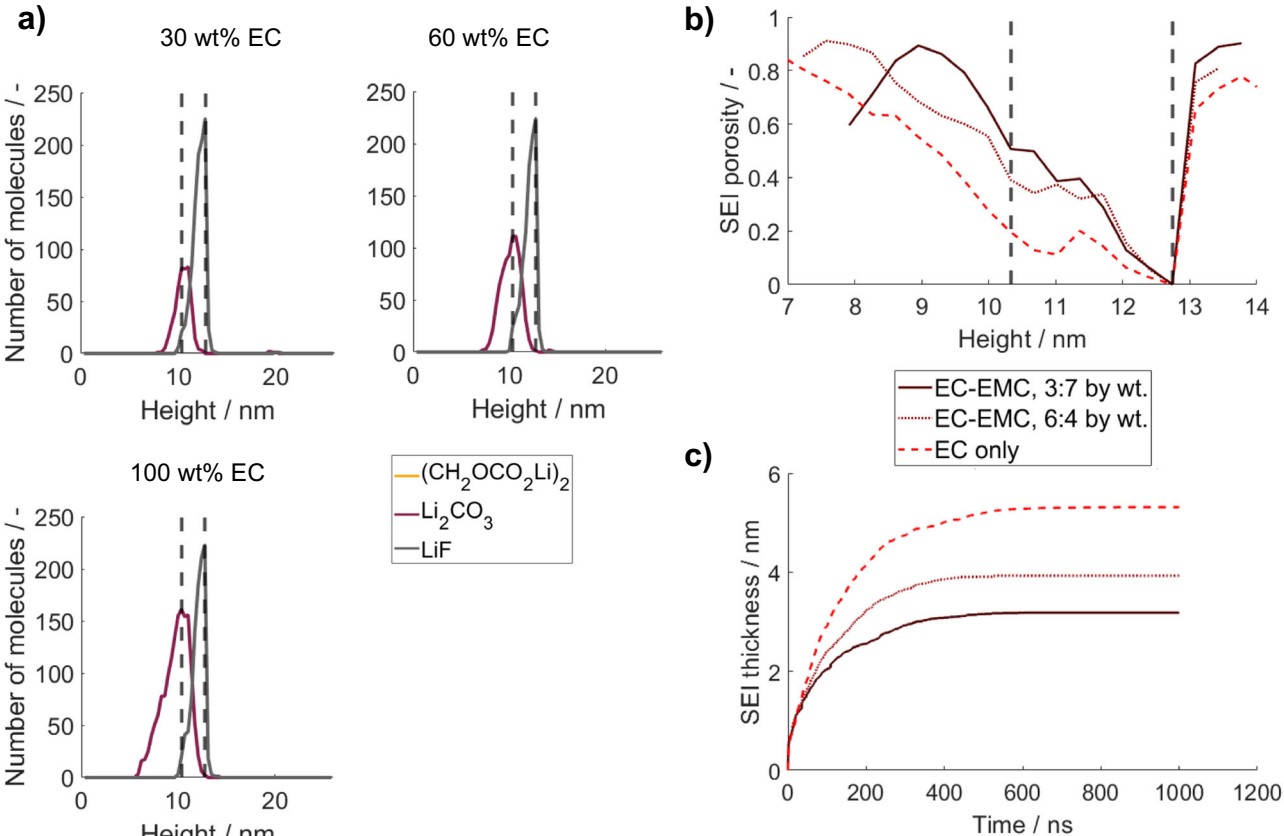

**Fig. 7 | Variation of the EC concentration in the solvent between 30 wt% and 100 wt% with 1.2 M LiPF$_6$. a** Number of SEI molecules after a simulated contact time of Li with the electrolyte of 1 μs over height. The dashed lines represent the initial lithium metal surface (left) and the maximum electron transport distance (right). **b** SEI porosity after 1 μs over height for the investigated EC weight fractions. **c** SEI thickness evolution for the investigated EC weight fractions within 1 μs. Source data are provided as a Source Data file.

In addition, it was demonstrated that the Li$^+$ concentration and solvation environment has a vast impact on the morphology of the formed SEI due to its stabilization effect on the EC solvent[28]. High Li$^+$ concentrations, which are related to slow EC degradation kinetics, lead to a layered inorganic SEI. In contrast, low local Li$^+$ concentrations relate to fast EC degradation kinetics resulting in a more mosaic-like inorganic SEI and a higher amount of organic SEI species. This finding explains parts of the ambivalent literature on the SEI structure. It further suggests the controlling of the local Li$^+$ concentration as an interesting new strategy in rational SEI tuning. Further analyzed options of SEI tuning were the variation of salt and solvent concentration. Thereby, in contrast to the solvent concentration, the increase of salt concentration leads to accelerated surface passivation and simultaneously increases the LiF content of the SEI. Hence, a variation of the salt concentration can be applied for a purposeful, advantageous design of the SEI. In order to reveal the relation between the predicted SEI structures and its performance in future the predicted structures could either be tested experimentally or by an additional performance model.

Overall, the presented modeling approach gives interesting new, detailed and quantitative insight into the SEI formation on the μs time scale, which is challenging to obtain by MD simulation or in situ characterization. It is suitable for studying the mesoscale processes, which bridge experimental studies and atomistic calculations. In future, this approach could be applied to different electrolyte systems in order to predict the influence of different solvents, salts, additives or artificial passivation layers on the SEI composition and morphology and derive model-based design recommendations. It could be further transferred to additional cell chemistries such as e.g., Na-based batteries. In combination with advanced machine-learning algorithms which predict reaction mechanisms along with kinetic data this could even be used as a tool for electrolyte screening.

## Methods

The multiscale methodology applied in this study consists of three main parts and is depicted in Fig. 1: First, a three-dimensional kMC model was established, which models the formation processes and the resulting chemical composition of the SEI. This stochastic model was, second, directly coupled with a continuum model to ensure global electroneutrality within the simulation box. Third, DFT calculations were used to identify the energetics of the SEI formation reactions, which were then used as input parameters for the higher-scale models. The mentioned kMC model is based on a prior work of our group[49]. Here we extended this model to improve the physical reliability in terms of electroneutrality, ionic conductivity of the SEI and solid-state behavior of the lithium metal electrode and incorporated it into a multiscale modeling framework. In the following section, we first introduce the kMC model before we describe the coupling to the continuum model. Afterward, we elaborate on the model initialization and give some details on the performed DFT calculations.

### KMC model

The comparatively high efficiency of the kMC modeling approach primarily originates from the simulations on a molecular instead of an atomic level. Thereby, rare events such as reactions or diffusions are considered while the vibrational motions of atoms are neglected[61]. All considered rare events must be provided as a model input along with their kinetic parameters. Based on this, transition rates for each process are calculated, and the algorithm then stochastically chooses the

process to be performed in a random fashion. In this work, we applied an algorithm based on the Variable Step Size Method[62,63]. We modified this algorithm with the structured list approach suggested by Schulze[64] to increase its efficiency further. Thereby, all possible transition rates are calculated at the start of the simulation and updated after every time step. The detailed algorithm can be found in section 2 in the Supplementary information.

Our overall kMC modeling idea is shown in the middle of Fig. 1. A section of 5.165 nm × 5.165 nm of the lithium metal surface in contact with the liquid electrolyte is represented in the kMC box with a height of 25.823 nm. This modeling domain is divided into a fixed three-dimensional cubic lattice. Each lattice site can either be vacant or occupied by a molecule or ion. In this way, the lattice predefines all possible configurations. Based on the occupied and vacant sites in the simulation output, the average SEI thickness $\overline{d_{SEI}}$ in the $z$-direction can be calculated as follows:

$$\overline{d_{SEI}} = \frac{1}{n_x \cdot n_y} \cdot \left( \left( \sum_{(x,y)} z_{\max,SEI}^{(x,y)} - z_{\min,SEI}^{(x,y)} \right) + \Delta L \right) \qquad (1)$$

Thereby, $z_{\max,SEI}^{(x,y)}$ and $z_{\min,SEI}^{(x,y)}$ represent the height of the lowest and highest layer in the kMC box occupied by a SEI species which are connected to other SEI species in $z$-direction, $n_x$ and $n_y$ stand for the number of lattice sites in $x$- and $y$-direction and $\Delta L$ describes the edge length of the single voxels which is set to 0.3443 nm[49].

Further, the boundary conditions of the simulation box are defined to be periodic at the lateral walls to allow for free diffusion of species. The top of the box is an open boundary which is connected with a bulk electrolyte phase of constant composition, and the bottom boundary is closed, so that no species can leave or enter the box through this wall.

For this study, we considered three different types of events: diffusion of species $i$ in the electrolyte with the rate $\Gamma_{D,i}$, reactions $j$ in the solid and liquid phase with the rate $\Gamma_{R,j}$ and clustering of SEI species $i_{SEI}$ to form larger agglomerates with the rate $\Gamma_{Cl,i_{SEI}}$. In the following sections, we present their implementation and the respective transition rates. Thereby, we focus on the advancements of the model in comparison with our previously published work[49]. A list of all model parameters can be found as Supplementary Table 1 in the Supplementary Information.

**Transport of species and charge.** The diffusion rate is represented by $\Gamma_{D,i}$ and is implemented for all liquid components. This includes electrolyte species and dissolved intermediate products of the SEI formation. In general, species can hop to vacant sites in their next neighborhood. In order to allow for Li$^+$ conduction of the SEI, Li$^+$ is additionally enabled to hop to the next neighbors, which are occupied by clustered SEI species. Next neighbors are connected with the current lattice site with their face (horizontal diffusion), their edge (diagonal diffusion), or their corner. Since diffusion in solid phases such as in lithium metal and in SEI clusters is comparatively slow, it is not expected to have a significant influence on the SEI composition and morphology and is therefore neglected. The respective transition rates for the diffusion in the liquid phase are calculated with the following equations, which are based on the derivation by Drews et al.[65]:

$$\Gamma_{D,i}^{face} = \frac{D_i}{2(\Delta L)^2} \qquad (2)$$

$$\Gamma_{D,i}^{edge} = \frac{D_i}{4(\Delta L)^2} \qquad (3)$$

$$\Gamma_{D,i}^{corner} = \frac{D_i}{6(\Delta L)^2} \qquad (4)$$

$D_i$ represents the macroscopic diffusion coefficient of species $i$. It is assumed to be the same constant value of $2.27 \cdot 10^{-10}\,\mathrm{m^2/s}$ for all considered species, which is in the range of previously reported values for EC systems[49,66]. Equation (2) is further applied to calculate the transition rate of the clustering processes of SEI species, which were introduced by Gerasimov et al.[49]. These clustering processes describe the precipitation of SEI components from the liquid phase on crystalline or amorphous solid SEI species of the same kind. The underlying driving force is that during the formation of the solid phase, a structural reorientation of neighboring SEI species reduces the total energy of the system, as observed in ref. 67.

The movement of charged species additionally depends on maintaining local electroneutrality by minimizing the local electrostatic forces introduced by the sum of charges in the system. In literature, the electrostatic forces in kMC models were e.g., calculated with the fast multipole method[68,69], Ewald Summation[70] or Cut-off methods[71,72]. However, these approaches are usually quite complex and computationally expensive, since the electric field has to be updated after every change of the charge configuration in the kMC box. To allow for reaching large simulation time scales, in this study we applied a simplified approach to calculate the electrostatic interactions. Thereby, we only consider the repelling forces based on the local charge distribution on the next neighbors of each charged species and assume that all charges behave like point charges. After the example of Pippig et al.[71], the charges are locally summed up with the electrostatic energy as follows:

$$E_n = \frac{1}{4 \cdot \pi \cdot \epsilon_0 \cdot \epsilon_R \cdot \Delta L} q_n \cdot \sum q_{nn} \qquad (5)$$

Here, $q_n$ represents the charge of the species on the current site $n$, and $q_{nn}$ stands for the charges of the species on the neighboring sites. Furthermore, $\epsilon_0$ and $\epsilon_R$ describe the vacuum permittivity and the relative permittivity, respectively. Based on this, the transport rate of charged species $\Gamma_{D,i_{ion}}$ is given by the applicable diffusion rate $\Gamma_{D,i}^x$, with $x \,\epsilon\, \{face, corner, edge\}$ (see Eqs. (2)–(4)) and, in case of repelling local forces, an exponential dependency on the local electrostatic energy as follows:

$$\Gamma_{D,i_{ion}} = \begin{cases} \Gamma_{D,i}^x \cdot \exp\left(\frac{E_n}{k_B \cdot T}\right) & \text{for } E_n > 0 \\ \Gamma_{D,i}^x & \text{for } E_n \leq 0 \end{cases} \qquad (6)$$

Thereby, $k_B$ represents the Boltzmann constant, and $T$ the temperature.

**Reactions.** The considered reaction network of SEI formation is summarized in Table 1 and graphically displayed in Supplementary Fig. 1. On the pure lithium metal surface, the lithium atom acts as electron donor, i.e., is oxidized, which promotes reductive electrolyte degradation. The rapid oxidation of Li is reflected in the low electrode potential and is the driving force for the rapid SEI formation on lithium metal. In this study, the electrolyte consists of EC + EMC and LiPF$_6$. Since the degradation of EMC on lithium metal is known to be significantly slower than EC or the conductive salt[73] and is too slow for the maximum simulation time of 1μs, the degradation of EMC is neglected in this study. In terms of the electrolyte solvent EC, the SEI species LiEDC and Li$_2$CO$_3$ are known as the main degradation products since the early work of Aurbach et al. in the 1990s[12,74,75]. The related reaction pathways were extensively studied by ab initio calculations[57,76]. The two pathways considered in this paper were extracted from the paper of Wang et al.[76] and were recently confirmed by the study of Spotte-Smith et al.[31]. Both pathways start with the electrochemical ring-opening of EC, denoted as R2. In a second reduction step, R4, one ethene can be released. The remaining carbonate ion undergoes the chemical reaction, R5, forming Li$_2$CO$_3$ with Li$^+$ from the solution.

Alternatively, two ring-opened LiEC can react in R3 to form the organic species LiEDC. Thereby, one ethene molecule is released. From literature we could not identify EC degradation pathways for $CO_2$ production which do not either require impurities such as water or HF[3,16] nor a direct adsorption of the solvent molecule with the lithium metal surface[32]. We therefore did not implement any $CO_2$-producing reaction in our study.

In the absence of water or other contaminants, LiF is considered to be the main degradation product of the conductive salt $LiPF_6$[32,75,77]. Based on ab initio MD calculations from the literature[32,78] we assume each salt structure to quickly release up to three fluoride anions by the electrochemical reactions R6-R8 which are summarized in Table 1. This means that from each salt anion close to the lithium surface, three LiF can be formed.

In the kMC model, one or two reactants can form up to two products per reaction process. Therefore, reactants have to be present on neighboring sites, as can be seen in the middle of Fig. 1, and products are placed on the reaction site or vacant next-neighbor sites. We assume that gaseous products are volatile, insoluble and are instantaneously transported away from the surface. Hence, ethene is not further considered in the simulation box after its production, and all related backward reactions are neglected. The transition rate of the chemical reaction process $j$ is calculated with an Arrhenius-type approach, as shown in Eqs. (7) and (8).

$$\Gamma_{R,j} = k_{0,j} \cdot \exp\left(-\frac{E_{A,j}}{R \cdot T}\right) \quad (7)$$

$$E_{A,j^{forw}} = \Delta G_j^{\ddagger} \quad (8)$$

$$E_{A,j^{back}} = \Delta G_j^{\ddagger} - \Delta_R G_j \quad (9)$$

$E_{A,j}$ represents the energy barrier defined by Eqs. (8) and (9) for forward and backward reactions, respectively. Thereby, $\Delta G_j^{\ddagger}$ represents the activation energy, and $\Delta_R G_j$ stands for the free energy of the respective reaction $j$. Those energy values for all reactions R1-R8 were extracted from DFT calculations for this study and are summarized in Table 1. Furthermore, $k_{0,j}$ stands for the frequency factor, which is usually in the range of $10^{12}$ to $10^{13}$ (see ref. 61), and was chosen to be $10^{13}$, corresponding roughly to kT/h, for all reaction processes in this study. Eventually, $R$ describes the ideal gas constant and $T$ the temperature, which is set to be 298.15 K in accordance with the DFT calculations throughout this study. The total activation energy of the oxidation of the only solid reactant lithium metal $E_{A,R_1^{forw}}^{tot}$ accounts in addition for the binding energy $E_A^{bond}$ between the lithium metal atoms:

$$E_{A,R_1^{forw}}^{tot} = E_{A,R_1^{forw}} + E_A^{bond} \cdot n_{nn}^{Li} \quad (10)$$

Here, $n_{nn}^{Li}$ stands for the number of directly neighbored lithium metal atoms. A similar approach was e.g., chosen by Callejas-Tovar et al.[79]

The transition rates of electrochemical reactions further depend on the potential difference between the electrode and liquid electrolyte $\Delta\Phi_{KMC}$. Here we follow the approach used for kMC by Röder et al.[50,80]:

$$\Gamma_{R,j_{ox}} = k_{0,j} \cdot \exp\left(-\frac{E_{A,j}}{R \cdot T}\right) \cdot \exp\left(\frac{\beta\Delta\Phi_{KMC}F}{RT}\right) \quad (11)$$

$$\Gamma_{R,j_{red}} = \sigma(z) \cdot k_{0,j} \cdot \exp\left(-\frac{E_{A,j}}{R \cdot T}\right) \cdot \exp\left(\frac{-(1-\beta)\Delta\Phi_{KMC}F}{RT}\right) \quad (12)$$

This Galvani potential is calculated by the continuum model and transferred to the kMC model as an input in each sequence.

Furthermore, $\beta$ stands for the symmetry factor and $F$ represents the Faraday constant.

Since the movement of single electrons is too fast to be directly considered in the kMC model, we defined the electron factor $\sigma(z)$ as a macroscopic approximation for the availability of electrons for reduction reactions (cf. Eq. (12)) depending on the distance $\Delta z = z - z_{Li,max}$ to the lithium metal electrode in $z$-direction:

$$\sigma(z) = \begin{cases} 1, & for\ z \leq z_{Li,max} + \Delta L \\ \exp\left(\frac{\ln(p_{el})}{z_{el,max}} \cdot \Delta z\right), & for\ z_{Li,max} + \Delta L < z \leq z_{Li,max} + \Delta L + z_{el,max} \\ 0, & for\ z > z_{Li,max} + \Delta L + z_{el,max} \end{cases} \quad (13)$$

Thereby, $p_{el}$ describes the electron probability at the maximum electron tunneling distance $z_{el,max}$ and $z_{Li,max}$ stands for the height of the kMC-box initially occupied by lithium metal. For this study, $p_{el}$ was chosen to be 0.01 according to ref. 49, and $z_{el,max}$ was set to be 2 nm which is a typical range for electron tunneling already used in previous simulation studies[81]. A detailed derivation of the electron factor is provided in section 3 of the Supplementary Information.

## Continuum model

Local electroneutrality was already accounted for in the kMC model by Eqs. (5) and (6). However, if used as a standalone model, this does not include any control of global electroneutrality. Hence, oxidation and reduction processes can occur independently without consideration of the availability of excess or lack of electrons. Therefore, we here couple a three-dimensional kMC model to the following macroscopic charge balance following the MPA2 coupling algorithm developed by Röder et al.[50]:

$$C^{DL}\frac{d\Delta\Phi_{Conti}}{dt} = I - r_{el} \cdot F \quad (14)$$

As shown in Fig. 1, we assume, that the potential drop $\Delta\Phi_{Conti}$ between the electrode and the liquid electrolyte behaves like a classical electrochemical double layer over the electrode/electrolyte interface with a capacity of $C^{DL}$, which is set to $0.2F/m^2$ [50]. Thereby, only one electron is transferred per electrochemical reaction step. Furthermore, $I$ represents the applied current, which is set to be 0 A in this study. $r_{el}$ refers to the total rate of electrochemical reactions in the system. It is calculated as the difference between the total reduction and oxidation rates $r_{el} = r_{red} - r_{ox}$ which depend on the output of the kMC model according to the following correlations:

$$r_{ox} = k_{ox} \cdot \exp\left(\frac{\beta\Delta\Phi_{Conti}F}{RT}\right) \quad (15)$$

$$r_{red} = k_{red} \cdot \exp\left(\frac{-(1-\beta)\Delta\Phi_{Conti}F}{RT}\right) \quad (16)$$

Here, the reaction constants $k_{ox}$ and $k_{red}$ are derived from the kMC output by Eqs. (17) and (18) for each sequence.

$$k_{ox} = \frac{\sum\Psi_{ox}}{t \cdot \Delta L^2 \cdot n_x \cdot n_y \cdot N_A} \cdot \frac{1}{\exp\left(\frac{\beta\Delta\phi_{KMC}(seq)F}{RT}\right)} \quad (17)$$

$$k_{red} = \frac{\sum\Psi_{red}}{t \cdot \Delta L^2 \cdot n_x \cdot n_y \cdot N_A} \cdot \frac{1}{\exp\left(\frac{-(1-\beta)\Delta\phi_{KMC}(seq)F}{RT}\right)} \quad (18)$$

In these equations, $\sum\Psi_{red/ox}$ describes the number of reduction/oxidation processes that were performed in the kMC model since time $t = 0$, $t$ is the time simulated by the kMC model, $n_x$ and $n_y$ are the

number of lattice sites in x and y direction, and $N_A$ stands for the Avogadro constant. The potential balance in Eq. (14) is solved in each continuum model sequence, which is called after every 1000 kMC loops, by using the ode15s solver of MATLAB. The resulting potential at the end of the sequence is fed back to the kMC model by $\Delta\Phi_{KMC}(seq) = \Delta\Phi_{Conti}(t^{seq})$. After computing the next 1000 kMC steps with this potential value, the final number of electrochemical reactions and the simulated time are again transferred to the continuum for determining the next potential.

### DFT calculations

The DFT evaluations were conducted with the Gaussian16 package. All the chemical structures were optimized to their local minimum at the B3PW91 level of theory with 6–311 G(3df) basis sets[82,83]. The free energy values were converted from the default 1 atm condition to the standard state of 1 M. An implicit solvation model based on density (SMD) was implemented to emulate the solvation environment in cyclic-carbonate-based electrolyte[84]. More details on the performed DFT calculations can be found in Supplementary Table 3. We found that the lithiation relaxation energies vary from species to species. To prevent this phenomenon from influencing the reaction thermochemistry calculation, each elementary reaction step was evaluated based on the electronic energy difference from transition state (TS)/product to reactant with proper thermal energy corrections to obtain the free energy barrier/change from the total electronic energy value of each species.

### Model initialization

For the initialization of our simulation box, we assume a clean and pristine lithium metal surface which comes into contact with the pure electrolyte solution without any contaminants being present in the electrode or electrolyte. The first 30 bottom layers of the kMC simulation box, starting from $z = 0$ and ending at $z_{Li,max}$, are initially completely filled with lithium metal atoms to represent the lithium metal electrode. The remaining lattice sites, i.e., $z > z_{Li,max}$, are randomly occupied by electrolyte molecules or vacant. Thereby, the number of species $n_i$ depends on the chosen salt concentration and solvent composition, which is 1.2 M of $LiPF_6$ in EC in the standard case and can be calculated by Eq. (19).

$$n_i = \left\lfloor N_A \cdot c_i \cdot \Delta L^3 \cdot n_x \cdot n_y \cdot \left(n_z - \frac{z_{Li,max}}{\Delta L}\right)\right\rfloor \text{ with } i \,\epsilon\,\{EC, PF_6^-, Li^+\} \quad (19)$$

The concentrations $c_i$ are varied for the modeling studies on the effect of salt and solvent concentrations. After placing the molecules, all possible events and transition rates are determined and saved in the structured lists according to Schulze et al.[64]. In the last step, the potential difference between the electrode and electrolyte for the first kMC sequence $\Delta\Phi_{KMC}(seq = 1) = \Delta\Phi_{Conti}(t = 0)$ is calculated by minimizing Eq. (20) in order to balance out the probability of oxidation and reduction processes. Thereby, the index $j_{el}$ stands for the implemented electrochemical reaction processes, and $l$ represents the lattice sites in the kMC box.

$$\min_{\Delta\Phi}\left(\left|\sum_l \sum_{j_{el}} \Gamma_{red}^{j_{el},l}(\Delta\Phi) - \sum_l \sum_{j_{el}} \Gamma_{ox}^{j_{el},l}(\Delta\Phi)\right|\right) \quad (20)$$

The presented multiscale model was implemented in MATLAB, and all simulations were performed using MATLAB Version 2021a. The calculations were performed on an i7-8700 CPU with 16 GB RAM.

### Data availability

Source data are provided with this paper in the KITopen repository[85] under https://doi.org/10.35097/1687.

### Code availability

The methods described in this article were implemented using custom Matlab (R2020a) code. Since a future commercialization and exploitation of the code is intended, the code is not publicly available. Further explanation of the methodology is available from the authors upon request.

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

## Acknowledgements

J.W. and U.K. acknowledge the financial support by the German Federal Ministry of Education and Research through funding the project "Lillint – Thermodynamic and kinetic stability of the Lithium-Liquid Electrolyte Interface" (03XP0225F), M.G. acknowledges the funding by the German Research Foundation (DFG) via "SiMET—Simulation of Mechano-Electro-Thermal Processes" (281041241). D.K. and P.B.B. acknowledge the Assistant Secretary for Energy Efficiency and Renewable Energy, Office of Vehicle Technologies of the US Department of Energy through the US-Germany Cooperation on Energy Storage under Contract DE-AC02-05CH11357. Computational resources from the Texas A&M University High-Performance Research Computing are gratefully acknowledged. We further acknowledge support by the KIT-Publication Fund of the Karlsruhe Institute of Technology.

## Author contributions

F.R., U.K., and P.B. conceived the idea and supervised the work. The multiscale model was developed by J.W. and M.G. D.K. performed the DFT calculations and discussed implementations in the multiscale model. The multiscale modeling studies were conducted by J.W. who also analyzed the data and wrote the manuscript. U.K. and P.B. revised the manuscript. All authors contributed to the scientific discussion of the manuscript.

## Funding

## Competing interests

The authors declare no competing interests.
