## [Peer Review File · Nature Communications]

REVIEWER COMMENTS

Reviewer #1 (Remarks to the Author):

In the article “Knowledge-driven design of Solid-Electrolyte Interphases on lithium metal: Insights from multiscale modelling”, the authors have developed a multiscale approach including KMC, DFT and continuum electroneutrality model to demonstrate the initial SEI formation on Li metal under open circuit conditions in the first microsecond and reveal its composition, thickness. The novelty of the article lies in being able to increase the simulation timespan from 1 ps to 1 μ s as compared to pure DFT or AIMD, and gain insights into the structure of the SEI with changes in salt and solvent concentrations. The article is decently written with good discussion of the results.

A few comments that need more clarification are outlined below in a section-by-section fashion:

- Introduction, Lines 116-117: The authors write “None of the previously mentioned models can yet model the SEI formation on lithium metal on a molecular resolution up to technically relevant time scales”. The authors have simulated the first μ s of SEI formation which is indeed a big leap from 1 ps but still much smaller than relevant battery timescales. A battery operating at 1C typically operates for an hour which is orders of magnitudes higher than 1 μ s. Battery calendar aging under open circuit (the electrochemical condition used in these simulations) is also generally done on the time scales of months. The authors are advised to reword the statement.

- Results , Table 1:

- o The reasoning for no LEDC formation in most of the simulation results is not evidently clear from the reaction mechanism. Reaction 2 for the ring opening is shown to be barrierless (0 kCal/mol) when the lithium ion is not coordinated to the EC. The authors posit most of their arguments of non-LEDC formation in their simulations on the high barrier of Li⁺ coordinated EC reacting to form LiEC (12.05 kCal/mol). In a typical Li-ion battery electrolyte, there should be plenty of free uncoordinated EC molecules alongside solvated LiEC⁺. This would imply that the reaction 2 to form LiEC should be favorable in the presence of a free Li⁺ (provided by the Li metal) and free EC molecule. So, if LiEC is favorably formed what is

stopping reaction 3 from happening where 2 LiEC's are combining to form LEDC?
Its barrier is 2.93 kcal/mol (R3) which is still less than the 3 kcal/mol used for the
PF6- reactions to form LiF (R6-R8).

o Furthermore, reaction 2 assumes 1 Li ion coordinated to 1 EC solvent molecule. In
literature (J. Phys. Chem. B 2015, 119, 1535–1545), lithium ions are generally
coordinated with multiple solvent molecules. Quoting from the article referenced:”
Li+ prefers a tetrahedrally coordinated first solvation shell regardless of which
species are involved, with the specific preferred solvation structure dependent on
the organic solvent”. Why is there a difference in the actual solvation structure (1
Li+ with 4 EC's) with the initial reactant given in R2 (1 Li+ with 1 EC)?

o Reaction 4 starts from the intermediate [LiEC]-

. There is no other reaction in the

table that forms this intermediate (R1-R8). How do reactions R4 and R5 occur if
the [LiEC]- is not present initially to begin with? Is there a reaction that is missing
from this table, or are the authors assuming an initial concentration of [LiEC]- in
the simulation.

o In a similar vein, on lines 299-302, the authors write “While in this work the
formation of inorganic carbonates was found to be more likely in an electron-rich
environment due to lower energy barriers of R3 compared to R4, the formation of
organic LiEDC was energetically preferred in the former study of Gerasimov et al”.
A look at the table shows that energy barrier for R3 is actually higher than R4 (2.93
kcal/mol vs 0 kcal/mol).

- Results Section, Lines 198-200: The authors write “The lithium metal anode provides a
large source of Li+, while the intercalated ions in graphite electrodes are not available for
reaction under open-circuit voltage conditions.” This is incorrect. Graphite based lithium
ion batteries have been proven to show Li inventory loss under calendar aging conditions
due to loss of intercalated Li to the SEI and is in fact the primary mode of degradation for
graphite-LFP cells (see Journal of Power Sources 208 (2012) 296–305). Quoting verbatim
from the referenced article's abstract: “The extent of capacity fade strongly increases with

storage temperature and to a lesser extent with the state of charge. From indepth data analysis, cyclable lithium loss was identified as the main source of capacity fade. This loss arises from side reactions taking place at the anode, e.g. solvent decomposition leading to the growth of the solid electrolyte interphase.”

- Results, Validation with MD model:

- o The authors validate their simulation results against reactive MD simulations with ReaxFF in literature for qualitative and quantitative comparison. This is interesting as KMC and MD are in this reviewer’s mind two very different numerical approaches and require much different inputs. Let me explain in detail. Lets say we are solving a simple heat equation by finite difference (FDM), finite volume (FVM) or finite element (FEM). The partial differential equation remains the same, the input properties remain the same, it is only the numerical solution approach that changes across FDM, FVM, FEM and hence we can compare these methods against each other. When it comes to comparing their KMC model with MD model, can the authors comment on if the input properties remain the same across both simulation paradigms? The authors have listed their input properties in Table S1. Will this remain the same as the MD simulations? Is their a unique set of inputs in the KMC model that will match the results of the MD model or you can vary some other property to get the same result? In this case the authors varied the barrier of R2 to get a match with the MD results; can the authors comment on its uniqueness or there can be other input properties (like let’s say the barrier of some other reaction) that can be varied to get a match with the MD data?

- o Is there any experimental validation available in literature of layering of the inorganic SEI seen in the authors simulations? Typically, experimental literature talks about layers in the sense of inorganic inner layer and organic outer layer but here the authors are talking about layering in the inorganics itself.

- Results, Line 262-265: The authors state “Moreover, it displays an essential difference between lithium metal and graphite anodes, which was already discussed by He et al.: As reduced lithium with a very low open circuit potential is present from time $t = 0$, the SEI

formation starts immediately after contact of the electrolyte with the anode material and does not require any external current". Can the authors clarify this more? Is the implication that graphite will not form an SEI when it is put in contact with electrolyte under open circuit conditions? Typically, SEI formation has both chemical and electrochemical nature (current/voltage driven) and chemical SEI can form irrespective of current in both graphite or Li.

- Results, Lines 342-350: This section talks about why Li_2CO_3 forms preferably over LEDC. It is confusing to the reviewer as pointed out in an earlier comment as well. Kindly reword it to make it more clear. Also, the way reactions are referenced here can be confusing. To the reader "...comparing the kinetics of the first and second reduction step", might be confused with the first and second reactions in the table when I believe it is referring to R2 and R4 respectively (correct me if I am wrong).
- Results, Influence of Salt and Solvent Concentration: This section is succinct and well explained.

Overall, this article presents interesting insights into the initial formation of the SEI. Of major note is the bilayer form of the inorganic SEI and influence of salt, solvent concentrations on SEI structure.

Reviewer #2 (Remarks to the Author):

In this work, the authors investigated the SEI formation on Li metal by using kMC/continuum simulations, which is an important issue for the cycling stability of Li metal anode in Li-ion batteries. However, some basic assumptions need to be confirmed. More deep analysis and explanations should be included. The detailed comments are described in the following :

(1) DFT calculations were used to determine the Gibbs free energies ΔR_G and activation energies ΔG^\ddagger of each reaction in Table 1. The energy barriers of these reactions were based on self-reaction. However, the reactions are occurred on the surface of Li metal, which must be affected by the Li metal. The ΔR_G and ΔG^\ddagger of these reactions should be calculated on Li surface.

(2) Considering the assumption of self-reaction, the particularity of metal Li was not reflected, i.e., the simulation results on SEI formation remain the same for any other cathode or anode surfaces if the ion concentration distribution is the same. Deep explanations should be provided.

(3) The ring-opening reaction of EC may involve dehydrogenation (Langmuir 2021, 37, 5252–5259). Whether the dehydrogenation of EC have been considered?

(4) Since the ion concentration distribution is the key point for a reliable SEI simulation, the scientific basis for the concentration distribution of species, such as Li ions and PF₆⁻, should be clarified.

(5) In order to enrich the readability of the manuscript, maybe you can review the more relevant references which introduce the multi-scale computation method for developing the second-battery materials, e.g., Chinese Physics B 25(1), 018212 (2016).

(6) The ethylene is one of the reaction products according to the reaction formula. However, ethylene molecules were not found in Figure 1 and Figure 3. In addition, the released ethene molecules compared with MD simulation were shown separately in Figure 2. But CO₂ products were also involved in MD simulation, why the product of CO₂ was not considered in kMC simulation?

Reviewer #3 (Remarks to the Author):

The authors present a multiscale kinetic Monte Carlo/continuum model to analyze the initial SEI formation when Li metal is in contact with the electrolyte. The activation energies of reactions are calculated from DFT simulations. The layered structure of SEI, the effects of the salt concentration and the EC concentration are studied in this manuscript. The writing of this manuscript conforms to the standard of the scientific paper. The authors walk the reader through assumption technical derivation and some test demonstrations. I praise the authors for focusing on this challenging problem and perhaps this work can be considered as a first step laying out a possible approach. However, some conclusions of the numerical simulations seem fairly trivial. And, the effects of EDL, which are one key point in metal-ion batteries, on the formation of SEI are ignored. Besides, I have some major concerns as follow:

(1) For the innovation of this method used in this manuscript, the method of KMC + DFT + analytical model is already used in the lithium metal batteries, such as Min Feng et al 2022 J. Electrochem. Soc. 169 090526. The authors should explain the differences and advantages of their method compared with other similar methods. Besides, the authors should add a more comprehensive review of the literature about the similar methods.

(2) At present, there are a large number of the experimental results of SEI. The authors should compare the simulation results with the experimental results in this manuscript to show the advantages of their

method. Note that the layered structure of SEI, the detailed component of SEI under different electrolytes, and the time evolution of SEI, which can give the readers a more intuitive understanding of the analysis of their method.

(3) In the part of introduction, the calculation method of the activation energy has some other new methods, such as the hybrid quantum-classical method established in the metal-ion batteries 10.1016/j.jpowsour.2023.232880, and the used machine learning method 10.1063/5.0096027. The authors should cover these works in the introduction. Besides, the effects of EDL on the formation of SEI should be analysed.

(4) In actual battery conditions, the surface morphology of the electrode is non-uniform, so the structure of SEI would be affected by the surface morphology. What is the effect of the surface morphology of lithium metal on the formation of SEI in their method? The advantages of this KMC coupled method can be reflected in the non-uniform initial design of the surface morphology.

(5) In the part of discussion, the authors have said their method has a 50 times larger time scale and a 32 times larger length scale than comparable ReaxFF MD simulations. I can not find a clear comparison of the calculated costs between their method and other calculations in this manuscript. The authors should state the calculation cost of their method. Besides, their model should compare not only with the ReaxFF MD simulations, but also with the similar coupled method mentioned in the comment (1).

Reviewer 1:

In the article “Knowledge-driven design of Solid-Electrolyte Interphases on lithium metal: Insights from multiscale modelling”, the authors have developed a multiscale approach including KMC, DFT and continuum electroneutrality model to demonstrate the initial SEI formation on Li metal under open circuit conditions in the first microsecond and reveal its composition, thickness. The novelty of the article lies in being able to increase the simulation timespan from 1 ps to 1 μ s as compared to pure DFT or AIMD, and gain insights into the structure of the SEI with changes in salt and solvent concentrations. The article is decently written with good discussion of the results. A few comments that need more clarification are outlined below in a section-by-section fashion:

We thank the reviewer for the positive feedback and careful evaluation of our manuscript. In the following we address each remark individually.

Introduction, Lines 116-117:

Remark 1.1: The authors write “None of the previously mentioned models can yet model the SEI formation on lithium metal on a molecular resolution up to technically relevant time scales”. The authors have simulated the first μ s of SEI formation which is indeed a big leap from 1 ps but still much smaller than relevant battery timescales. A battery operating at 1C typically operates for an hour which is orders of magnitudes higher than 1 μ s. Battery calendar aging under open circuit (the electrochemical condition used in these simulations) is also generally done on the time scales of months. The authors are advised to reword the statement.

The authors agree that technically relevant time scales in terms of calendric aging or cell cycling are in the order of seconds to month which is not reached with the presented method, yet. However, focus of this research was the initial SEI formation on Li metal, which is within milliseconds to minutes. Here, we significantly increased the investigated time scale while keeping a molecular resolution. This allows us to identify important mesoscale system dynamics such as reaction- and transport limitations of different processes during the formation.

Changes made: For clarification we rephrased the sentence on page 3, lines 119 and 120 as follows: “None of the previously mentioned models can yet model the SEI formation on lithium metal on a molecular resolution **above the nanosecond time scale.**”

Results, Table 1:

Remark 1.2: The reasoning for no LEDC formation in most of the simulation results is not evidently clear from the reaction mechanism. Reaction 2 for the ring opening is shown to be barrierless (0 kCal/mol) when the lithium ion is not coordinated to the EC. The authors posit most of their arguments of non-LEDC formation in their simulations on the high barrier of Li⁺ coordinated EC reacting to form LiEC (12.05 kcal/mol). In a typical Li-ion battery electrolyte, there should be plenty of free uncoordinated EC molecules alongside solvated LiEC⁺. This would imply that the reaction 2 to form LiEC should be favorable in the presence of a free Li⁺ (provided by the Li metal) and free EC molecule. So, if LiEC is favorably formed what is stopping reaction 3 from happening where 2 LiEC's are

combining to form LEDC? Its barrier is 2.93 kcal/mol (R3) which is still less than the 3 kcal/mol used for the PF6⁻ reactions to form LiF (R6-R8).

We agree that in a normal Li-ion battery there should be uncoordinated EC available. However, here we analyse the electrolyte decomposition directly on or close to a lithium metal surface, which can be considered as an almost infinite source of Li, which readily oxidizes and forms Li⁺-ions and a double layer at the interface when it comes into contact with the liquid electrolyte. Thereby, it provides a high concentration of Li⁺-ions close to the lithium metal surface. This high concentration makes it very likely that the EC molecules in this area are coordinated with a Li⁺-ion, which would lead to the higher ring-opening barrier. Since the exact coordination in this interface region is very complex and constantly changing, we varied the energy barrier of the first EC reduction step (R2), representing different ratios of coordinated/free EC in Figure 1 and Figure 2. We then compared the results with ReaxFF MD simulations from literature (J. Mater. Chem. A 8, 17036–17055 (2020)) and found that a high energy barrier for EC reduction agrees best with both, the reported amount of ethene production and the observed SEI structure after 20 ns.

The LEDC-forming reaction (R3) is further competing with the second EC reduction step (R4), which has an energy barrier of 0 kcal/mol. This means, as long as electrons are sufficiently available and the first reduction step is slow, the fast reduction of LiEC to LiCO₃⁻ (R4) is preferred over the slower LiEDC formation (R3). This is especially valid close to the lithium metal surface where the availability of free electrons is very high. And it decreases starkly with the distance to the surface. For more details also see our response to Remark 1.10.

Changes made: We rephrased the following paragraph on page 12, lines 372 – 380 as follows in order to clarify our argumentation:

“The reason why Li₂CO₃ is preferably produced over LiEDC can be explained by comparing the kinetics of both EC degradation pathways in R2-R4. As previously pointed out by Yu et al. [58], the second EC reduction step R4 which reduces LiEC to LiCO₃⁻ is faster than the first EC reduction step R2, which reduces EC to the ring-opened LiEC. This is also reflected in our kinetic parameters (cf. Table 1), in which the activation barrier of R2 is 12.05 kcal/mol, and the barrier for R4 is 0 kcal/mol. Hence, as long as electrons are readily available, which they are close to the Li surface, newly formed LiEC can quickly undergo a subsequent reduction reaction via reaction R4, leading to low LiEC concentrations and significant LiCO₃⁻ production. In contrast, two LiEC at neighboring sites are needed to produce LiEDC (cf. R3). As the LiEC concentration is low, this makes LiEDC production unlikely close to the surface.”

Remark 1.3: Furthermore, reaction 2 assumes 1 Li ion coordinated to 1 EC solvent molecule. In literature (J. Phys. Chem. B 2015, 119, 1535–1545), lithium ions are generally coordinated with multiple solvent molecules. Quoting from the article referenced:” Li⁺ prefers a tetrahedrally coordinated first solvation shell regardless of which species are involved, with the specific preferred solvation structure dependent on the organic solvent”. Why is there a difference in the actual solvation structure (1 Li⁺ with 4 EC’s) with the initial reactant given in R2 (1 Li⁺ with 1 EC)?

We thank the reviewer for bringing up this important point. We previously studied the effect of the lithium coordination Li(EC)_n (n=1...4) in our 2002 JACS paper (J. Am. Chem. Soc. 2002, 124, 4408-4421). Thereby, we found that the changes in the activation barriers for ring opening, and other properties are not significant when changing between coordination number 1 to 4. For reference

please see table 9 from the referred paper which we reproduce below. The important entries are highlighted in yellow:

Table 9. Comparisons between Bond Lengths ($R/\text{\AA}$ at B3PW91/6-31G(d)), Binding Energies per Solvent Molecule ($BE/\text{kcal/mol}$ at B3PW91/6-31G(d)), Adiabatic Electron Affinity ($EA/\text{kcal/mol}$), Ring-Opening Barriers ($E_a/\text{kcal/mol}$), Releasing Energies (RE) of Radical Anion Formation for $\text{Li}^+(\text{EC})_n$ ($n = 1-4$), and $(\text{EC})_n\text{Li}^+(\text{VC})$ ($n = 0-3$)

structures	R	BE	EA^b	E_a^c	RE^d
$\text{Li}^+(\text{EC})$	1.764	50.3	92.5/45.9	11.5/9.6	121.7/75.8
$\text{Li}^+(\text{EC})_2$	1.814	44.7	72.4/45.2	11.0/10.5	102.3/73.3
$\text{Li}^+(\text{EC})_3$	1.893	37.8	64.6	11.1	93.9
$\text{Li}^+(\text{EC})_4$	1.965	32.2	59.0	10.2	86.9
$\text{Li}^+(\text{VC})$	1.773	46.4	97.0/50.9	20.6/21.2	110.7/64.3
$(\text{EC})\text{Li}^+(\text{VC})$	1.813, 1.825	43.1	77.4 (74.3)/52.1 (45.9)	11.1/10.0 20.4/22.6 13.5/17.8	104.2/74.8 91.8/63.6
$(\text{EC})_2\text{Li}^+(\text{VC})$	1.888, 1.903	37.0	70.1 (64.0)	8.8 20.1 17.2	92.2 81.4
$(\text{EC})_3\text{Li}^+(\text{VC})$	1.956 ^e , 1.981	31.7	64.1 (58.8)	10.1 21.1 18.4	85.4 77.0

^a Average of the three $\text{Li}^+\cdots\text{O}$ interactions. ^b The data outside and in parentheses for $(\text{EC})_n\text{Li}^+(\text{VC})$ correspond to the reductions of VC and EC, respectively. The data before and after slants refer to without and with CPCM corrections, respectively. ^c The first two data for $(\text{EC})_n\text{Li}^+(\text{VC})$ correspond to the ring-opening of EC and VC in their respective reduction-intermediate; the third one corresponds to the ring-opening of EC in the VC-reduction intermediate. ^d The two data for $(\text{EC})_n\text{Li}^+(\text{VC})$ correspond to the ring-opening of EC and VC; the results of $\text{Li}^+(\text{EC})_n$ from ref 21.

Changes made: We added the information to the manuscript, that the ring-opening energy is not significantly influenced by the exact ratio of coordination of Li^+ and EC and rephrased the concern paragraph on page 5, lines 156 – 163 as follows:

“It was recently shown by Kuai et al.²⁸, that the activation energy of this process is significantly impacted by lithium coordination. When coordinated with one Li^+ ion, the energy barrier was calculated to be 12.05 kcal/mol. As previously shown in literature⁵¹ this barrier is mostly independent of the exact coordination ratio of EC and Li^+ , and also applies if several EC molecules are coordinated with one Li^+ -ion. However, Kuai et al.²⁸ showed that the ring-opening barrier is significantly different for uncoordinated EC and becomes even negative. Similar “declining” energy surfaces are found in reactions R4 and R5. To transcribe this into the kinetic information, we assume the activation energies ΔG^\ddagger of the corresponding reactions to be 0 kcal/mol.”

We further rephrased the following sentences on page 6, lines 171 – 173 and lines 201 – 203 to clarify the meaning of the variation of ΔG^\ddagger of the EC reduction reaction R2: “Thereby, each parameter set represents a different stochastic average of the availability of uncoordinated vs. Li^+ -coordinated EC molecules close to the reaction site.”

“In contrast, a low Li^+ concentration increases the availability of uncoordinated EC molecules and thus – due to the negligible barrier for uncoordinated EC – facilitates EC degradation.”

Remark 1.4: Reaction 4 starts from the intermediate $[\text{LiEC}]^-$. There is no other reaction in the table that forms this intermediate (R1-R8). How do reactions R4 and R5 occur if the $[\text{LiEC}]^-$ is not present initially to begin with? Is there a reaction that is missing from this table, or are the authors assuming an initial concentration of $[\text{LiEC}]^-$ in the simulation.

We thank the reviewer for identifying this inaccuracy in our reaction table. Reaction 4 is meant to be the second reduction step of EC. In this step, LiEC is reduced and forms a carbonate species while releasing ethene. However, we see that our previous notation might lead to misunderstandings.

Changes made: For clarification we adjusted the chemical structures of reaction 4 in Table 1 as follows:

Remark 1.5: In a similar vein, on lines 299-302, the authors write “While in this work the formation of inorganic carbonates was found to be more likely in an electron-rich environment due to lower energy barriers of R3 compared to R4, the formation of organic LiEDC was energetically preferred in the former study of Gerasimov et al”. A look at the table shows that energy barrier for R3 is actually higher than R4 (2.93 kcal/mol vs 0 kcal/mol).

We thank the reviewer for noticing this spelling error. Indeed, it should be “due to lower energy barriers of the carbonate-forming reaction R4 compared to the organic-forming reaction R3”.

Changes made: *We corrected the sentence in the manuscript (p. 10, line 329 – 330) accordingly and added the highlighted details about the referred reactions in order to further improve the readability of our manuscript.*

Results Section, Lines 198-200:

Remark 1.6: The authors write “The lithium metal anode provides a large source of Li⁺, while the intercalated ions in graphite electrodes are not available for reaction under open-circuit voltage conditions.” This is incorrect. Graphite based lithium ion batteries have been proven to show Li inventory loss under calendar aging conditions due to loss of intercalated Li to the SEI and is in fact the primary mode of degradation for graphite-LFP cells (see Journal of Power Sources 208 (2012) 296–305). Quoting verbatim from the referenced article’s abstract: “The extent of capacity fade strongly increases with storage temperature and to a lesser extent with the state of charge. From indepth data analysis, cyclable lithium loss was identified as the main source of capacity fade. This loss arises from side reactions taking place at the anode, e.g. solvent decomposition leading to the growth of the solid electrolyte interphase.”

We agree that our previous statement on Li⁺-ion availability was too definite. What is meant is that the likelihood for Li⁺-ion availability under open-circuit conditions on a lithium anode compared to a graphite anode is significantly larger. This is reflected by multiple things: First of all, the Open Circuit potential (OCP) of graphite anodes always remains above the OCP of lithium metal, even in a fully charged state [Phys. Rev. B, Condens. Matter, 44, 17, 9170-9177 (1991)]. Second, the fully charged state of a graphite anode can only be reached by a first charging cycle which leads to a first SEI formation [Carbon, 105, 52-76 (2016), Solid State Ionics 148, 405-416 (2002)]. Hence, there will always be a first charging-induced passivation layer on lithium-containing graphite anodes, which is not present on pure lithium metal, which is investigated in our study. Consequently, the initial SEI formation on graphite electrodes takes only place during the first operation, which takes minutes to hours. Subsequent calendric ageing on graphite cells proceeds on a time scale of several months as also reported in the referred paper [J. Power Sources 208 (2012) 296–305], while the SEI formation on lithium metal which we report on occurs on the μs time-scale. Therefore, we think that our intended statement about the different conditions on lithium and graphite interfaces is still valid.

Changes made: To be more precise we rephrased the sentence on page 7, lines 209 – 212 as follows:

“The lithium metal anode provides a large source of Li⁺-ions via oxidation, while the intercalated ions in graphite electrodes show a comparatively lower availability under open-circuit voltage conditions due to a higher Open Circuit Potential⁶⁴ and charging-induced passivation layers⁵⁵.”

Results, Validation with MD model:

Remark 1.7: The authors validate their simulation results against reactive MD simulations with ReaxFF in literature for qualitative and quantitative comparison. This is interesting as KMC and MD are in this reviewer’s mind two very different numerical approaches and require much different inputs.

The reviewer is right that ReaxFF MD and kMC are two very different numerical approaches. While the MD method solves the Newton’s equations of motion based on interatomic interactions of the participant species, kMC is a stochastic modelling approach based on pre-defined transition rates which we derive from DFT calculations. This is indeed exactly what makes the comparison between the results so interesting.

Let me explain in detail. Lets say we are solving a simple heat equation by finite difference (FDM), finite volume (FVM) or finite element (FEM). The partial differential equation remains the same, the input properties remain the same, it is only the numerical solution approach that changes across FDM, FVM, FEM and hence we can compare these methods against each other. When it comes to comparing their KMC model with MD model, can the authors comment on if the input properties remain the same across both simulation paradigms? The authors have listed their input properties in Table S1. Will this remain the same as the MD simulations?

As the underlying equations are different, the models require also partly different input parameters. The deterministic ReaxFF MD model does not require the definition of pre-defined rates, but calculates the rate of diffusion of each particle by solving the Newton’s equations of motion and the rates of reactions by applying the ReaxFF. Therefore, parameters from Table S1 that go into the kMC rate calculations such as e.g. diffusion coefficient, frequency factor or distance of surface sites, are not needed in MD. Yet, in MD and kMC, the input conditions such as the system configuration, including pure lithium metal surface, the electrolyte composition and temperature were the same. Moreover, we compare the results of both simulation approaches on the same box size and after the same simulated time. Although both approaches are so different, they represent the same physical situation, and we find very clear similarities in the results of both methodologies (Figure S.1). As we see this good match as a validation for the kMC, we apply the more efficient kMC algorithm to investigate the systems dynamics on unprecedented larger length- and time scales.

Changes made: In order to point out the advantages of the comparison between kMC and MD results we added the following sentence to our manuscript on page 7 lines 224 – 227: *“Since MD and kMC are very different simulation paradigms, which were applied to the same chemical system with identical size, time and temperature, the comparison is a good approach for validating and benchmarking our modeling approach.”*

Is there a unique set of inputs in the KMC model that will match the results of the MD model or you can vary some other property to get the same result? In this case the authors varied the barrier of R2 to get a match with the MD results; can the authors comment on its uniqueness or there can be

other input properties (like let's say the barrier of some other reaction) that can be varied to get a match with the MD data?

All kinetic parameters were taken from DFT simulations. Thereby, all reaction energies except of the EC ring-opening barrier could be uniquely defined and DFT predicted different values for coordinated and uncoordinated EC. Further, since the number of produced ethene molecules is directly connected with the amount of degraded EC molecules it can only be influenced by EC-related processes. Hence, in order to identify the uncertain parameter, we chose the ethene production over time for the quantitative comparison between the MD and kMC model in Figure 2. Since the second-step EC-degradation reactions R3 and R4 are comparatively fast, the rate limiting process is the first EC-degradation step in reaction R2. This is, why we do not expect a similar effect on the kMC results if we vary a system parameter which is related to another process. In any case, as mentioned above, the energy barrier of R2 is the only uncertain parameter as only EC may come coordinated or uncoordinated.

Remark 1.8: Is there any experimental validation available in literature of layering of the inorganic SEI seen in the authors simulations? Typically, experimental literature talks about layers in the sense of inorganic inner layer and organic outer layer but here the authors are talking about layering in the inorganics itself.

To the best of our knowledge, there is so far no experimental study that reports on the layering of the inorganic SEI on lithium metal, as we report here. This might be due to the difficulty to perform precise experimental measurements on the nm-length scale. Cryo-TEM measurements which were reported by Xu et al. [Nano Letters, 20, 1, 418-425 (2020)] could identify SEI morphologies on the nm-length scale on electrochemically deposited Lithium but were not applied to the initial SEI on lithium metal. Another very recent study [JPS, 549, 232118, (2022)] highlighted the importance of native passivation layers which always cover the lithium metal and have a substantial influence on the initial SEI formation. With FIB-SEM measurements they could show a layered initial SEI with a higher fluorine signal in the inner and a higher oxygen signal in the outer layer in a similar electrolyte. However, they trace this back to an inorganic/organic layer and do not reach a nm-resolution with their measurement technique.

In contrast to experimental evidence, we would like to mention that a layered inorganic phase of the SEI has indeed been previously indicated by the following ReaxFF MD study [J. Mater. Chem. A, 8, 17036-17055 (2020)], which we use for validating our simulation results.

Changes made: In order to relate our research more clearly to literature and highlight the novelty of our simulation results, we added the following sentences on page 9, lines 300 – 304:

“Detailed experimental studies on the initial SEI formation on lithium metal and its resulting composition, especially with a sub- μm resolution, are scarce in literature^{4,54}. To the best of our knowledge, there is no experimental study, yet, which was able to reveal the here observed layering of the inorganic SEI. Future advancements in experimental methods may allow to reach a similar resolution and thus provide an experimental validation.”

Results, Line 262-265:

Remark 1.9: The authors state “Moreover, it displays an essential difference between lithium metal and graphite anodes, which was already discussed by He et al.: As reduced lithium with a very low open circuit potential is present from time $t = 0$, the SEI formation starts immediately after contact of

the electrolyte with the anode material and does not require any external current". Can the authors clarify this more? Is the implication that graphite will not form an SEI when it is put in contact with electrolyte under open circuit conditions? Typically, SEI formation has both chemical and electrochemical nature (current/voltage driven) and chemical SEI can form irrespective of current in both graphite or Li.

The cited statement aimed to highlight the reason for the observed very fast initial SEI formation on metal which is due to high reactivity of lithium metal. The intention was not to deny SEI formation under OCV conditions on graphite, once the graphite has been charged, i.e. lithiated to a certain amount. However, the time scale of this process is very different on both anodes: While we report on initial SEI formation in the order of ns to μ s, the initial SEI formation on graphite anodes requires a charging process where potential gradually drops below a certain potential, which usually takes several minutes. The subsequent SEI growth under calendric ageing conditions would be significantly slower and would take up to several months as shown in Journal of Power Sources 208 (2012) 296–305 which you cited in your previous remark.

Changes made: In order to prevent any confusion, we rephrased the statement and removed the comparison with graphite anode, since this is not the focus in this paragraph. The revised statement on page 9, lines 281 – 284 reads as follows:

"This shows the high reactivity of pure lithium metal, which was already described by He et al.4: As reduced lithium with a very low open circuit potential is present from time $t = 0$, the SEI formation starts immediately after contact of the electrolyte with the anode material and does not require any external current."

Results, Lines 342-350:

Remark 1.10: This section talks about why Li_2CO_3 forms preferably over LEDC. It is confusing to the reviewer as pointed out in an earlier comment as well. Kindly reword it to make it more clear. Also, the way reactions are referenced here can be confusing. To the reader "...comparing the kinetics of the first and second reduction step", might be confused with the first and second reactions in the table when I believe it is referring to R2 and R4 respectively (correct me if I am wrong).

Thank you for pointing out the vague argumentation. The main reason that prevents LEDC formation (R3) is, that it requires two LiEC to be present on neighbouring sites. Hence, a certain accumulation of this intermediate species would be required. However, since the first reduction step (R2) which produces LiEC is comparatively slow, whereas the second reduction step (R4), which consumes LiEC and produces carbonate, is comparatively fast, we do not get sufficient accumulation of this species in our system. Since both reactions are reduction processes, the prevalence of R4 over R3 (LEDC production) only holds as long as sufficient electrons or reductive species are available.

Changes made: We rephrased the paragraph on page 12, lines 372 – 380 as follows in order to clarify our argumentation:

"The reason why Li_2CO_3 is preferably produced over LiEDC can be explained by comparing the kinetics of both EC degradation pathways in R2-R4. As previously pointed out by Yu et al. [58], the second EC reduction step R4 which reduces LiEC to LiCO_3^- is faster than the first EC reduction step R2, which reduces EC to the ring-opened LiEC. This is also reflected in our kinetic parameters (cf. Table 1), in which the activation barrier of R2 is 12.05 kcal/mol, and the barrier for R4 is 0 kcal/mol. Hence, as long as electrons are readily available, which they are close to the Li surface,

newly formed LiEC can quickly undergo a subsequent reduction reaction via reaction R4, leading to low LiEC concentrations and significant LiCO_3^- production. In contrast, two LiEC at neighboring sites are needed to produce LiEDC (cf. R3). As the LiEC concentration is low, this makes LiEDC production unlikely close to the surface.

Results, Influence of Salt and Solvent Concentration:

This section is succinct and well explained.

Overall, this article presents interesting insights into the initial formation of the SEI. Of major note is the bilayer form of the inorganic SEI and influence of salt, solvent concentrations on SEI structure.

We once again thank the reviewer for the thorough evaluation of our manuscript as well as for the valuable comments and the overall positive feedback on our work. We are confident that the changes made and additional explanations provided enhance the understandability and readability of our manuscript.

Reviewer 2:

In this work, the authors investigated the SEI formation on Li metal by using kMC/continuum simulations, which is an important issue for the cycling stability of Li metal anode in Li-ion batteries. However, some basic assumptions need to be confirmed. More deep analysis and explanations should be included. The detailed comments are described in the following:

First of all, we would like to thank the reviewer for the thorough evaluation of our manuscript and for the important annotations. In the following we individually comment on each remark and are confident that this helps to further clarify and improve our argumentation in the manuscript.

Remark 2.1: DFT calculations were used to determine the Gibbs free energies ΔR_G and activation energies ΔG^\ddagger of each reaction in Table 1. The energy barriers of these reactions were based on self-reaction. However, the reactions are occurred on the surface of Li metal, which must be affected by the Li metal. The ΔR_G and ΔG^\ddagger of these reactions should be calculated on Li surface.

We agree that this is an important point which might need further explanation: It is important to note that all reactions in Table 1 are electron transfer reactions. Adsorption reactions were not directly considered in this study for the following reason: Per AIMD results we observed in previous publication (ACS Appl. Mater. Interfaces 2022, 14, 2817–2824), that the direct contact of EC/VC with lithium metal surface results in highly fragmented species such as ethene and/or carbon dioxide. These corresponding products due to “adsorption” cannot trigger the subsequent reactions. Since the lithium metal is highly reactive, it becomes rapidly oxidized, leading to solvated ions and the loss of a distinct lithium metal surface on which electrolyte molecules could adsorb. The PF_6^- degradation is a similar case as studied in another paper of ours (J. Phys. Chem. C 2023, 127, 1744–1751).

Changes made: *We added the following additional explanation on page 5, lines 144 – 147: “Thereby, we focus on electron transport reactions instead of adsorption reactions. This is due to the high reactivity of lithium metal which quickly leads to lithium oxidation and the loss of a distinct lithium metal surface on which electrolyte molecules could adsorb.”*

Remark 2.2: Considering the assumption of self-reaction, the particularity of metal Li was not reflected, i.e., the simulation results on SEI formation remain the same for any other cathode or anode surfaces if the ion concentration distribution is the same. Deep explanations should be provided.

The lithium metal anode influences our simulation results in several ways. First of all, the main driving force for SEI formation in our simulations is the spontaneous oxidation of Li with the electrolyte reducing. Therefore, the observed SEI formation is due not only to the electrolyte reduction but also to the Li oxidation reaction and complexation with the reduced electrolyte as shown in the simulations. Moreover, the anode material will determine the type of SEI products formed. In the case of Li, there is a significant contribution of Li surface atoms to the reaction. Such reactive surface atoms are not present in SEIs formed on intercalated graphite or on Cu surfaces, also because they show a different potential. Last but not least, the initial configuration of KMC contains Li metal with ~10 nm thickness. The electron transport probability with respect to metal surface is included in the model, which stands in profound differences compared to other materials, which would lead to different electron transfer rates.

Changes made: In order to further clarify the effect of the lithium metal anode on the simulation results we highlighted that the lithium metal oxidation is the driving force of SEI formation in the “Methods” chapter on page 20, lines 600 – 603 as follows: “On the pure lithium metal surface, the lithium atom acts as electron donor, i.e. is oxidized, which promotes reductive electrolyte degradation **The rapid oxidation of Li is reflected in the low anode potential and is the driving force for the rapid SEI formation on lithium metal.**”

Remark 2.3: The ring-opening reaction of EC may involve dehydrogenation (Langmuir 2021, 37, 5252–5259). Whether the dehydrogenation of EC have been considered?

The 2021 Langmuir paper refers to electrolyte oxidation, where dehydrogenation of EC takes place due to an oxidation reaction on LMO cathode surface. Here we discuss electrolyte reduction, which is a different reaction.

Remark 2.4: Since the ion concentration distribution is the key point for a reliable SEI simulation, the scientific basis for the concentration distribution of species, such as Li ions and PF₆⁻, should be clarified.

The movement of species in the electrolyte has been described in the methods section, as well as the initial placement of the species. We here give a short summary: In the initial configuration of the kMC box, which is shown on the right-hand side, the 30 bottom layers are completely filled with reduced lithium metal to mimic the lithium metal anode. The free space above is randomly filled with electrolyte species, namely EC, Li⁺ and PF₆⁻. Thereby, the number of molecules of each species is calculated based on the volume of the free space and the individual concentration of the respective molecules in the electrolyte (cf. Equation 19). During simulation, all non-solid species can move via diffusion (cf. Equations 2-4). In order to account for local electroneutrality in our system, the transport rates of charged species are calculated based on the local charge distribution in the system (cf. equation 5-6). Overall, this ensures a meaningful species distribution in the electrolyte in the box throughout the simulation.

Changes made: We added in the main text on page 6, lines 168 – 170 a link to the detailed explanation in the methods section as follows: “Details on the model, including the main processes of reaction, diffusion and clustering, and the initial homogeneous distribution of liquid species in the electrolyte can be found in the methods section. For the parameter study, we...”

Remark 2.5: In order to enrich the readability of the manuscript, maybe you can review the more relevant references which introduce the multi-scale computation method for developing the second-battery materials, e.g., Chinese Physics B 25(1), 018212 (2016).

We thank the reviewer for their advice. We reviewed the suggested paper and found that it gives a good overview on multiscale modelling techniques in Li-ion battery applications.

Changes made: We decided to cite the review paper in our introduction on page 2, line 76 as follows: “Theoretical calculations allow a complementary and detailed resolution of the governing processes inside a battery cell, enabling a targeted virtual optimization or tracking of the state of the battery and its components ranging from the atomistic to the macroscopic scale [18,19]”

Remark 2.6: The ethylene is one of the reaction products according to the reaction formula. However, ethylene molecules were not found in Figure 1 and Figure 3. In addition, the released ethene molecules compared with MD simulation were shown separately in Figure 2. But CO₂ products were also involved in MD simulation, why the product of CO₂ was not considered in kMC simulation?

Thank you for pointing out that kMC simulation of ethene was not very clear. As mentioned in the method section in the manuscript on page 21, lines 627 – 630, “We assume that gaseous products are volatile, insoluble and are instantaneously transported away from the surface. Hence, ethene is not further considered in the simulation box after its production, and all related backward reactions are neglected.” It is not explicitly tracked and can therefore neither be found as a reaction product in Figure 1 nor in Figure 3. However, since we track the number of occurred reactions, we can quantify the amount of released ethene molecules. This number is plotted in Figure 2 against the MD results.

Changes made: To make the calculation of gases clearer, we extended the caption of Figure 2 as follows: “The number of ethene molecules is calculated from occurrence of the gas-forming reactions R3 and R4 and ...”. Further we added the following explanation on page 8, lines 245 – 246: “Since it can only be produced as a side product of the EC degradation, it is directly related to the number of reduced EC molecules and can be quantified by the tracked number of the ethene forming reactions R3 and R4.”

Regarding CO₂: A characteristic of kMC calculations is, that the considered reaction pathways must be provided as a model input. In this study, we restricted our calculations to the most commonly reported EC degradation reactions, namely the formation of Li₂CO₃ and of organic LiEDC. Both release one ethene molecule, each. EC-degradation reactions that might lead to CO₂ production either require impurities such as water or HF ([Energy Environ. Sci. 14, 10, 5289-5314 (2021), Solid State Ion. 148, 3-4, 405-416 (2002)], which are neglected in our study, or the reactions are only scarcely reported, i.e. are not certain, and therefore not considered. The CO₂ production in the MD study can most probably be traced back to the initial fragmentation of EC molecules that directly adsorbed at the lithium metal surface before any passivation occurred. Since the direct contact between solvent

and lithium is quickly prevented by a first passivation layer, we did not consider these processes in our mesoscale kMC/continuum study. However, if CO₂-producing reaction pathways with reasonable reaction energies can be found, this would be certainly of interest for future studies.

Changes made: We added the following explanation to our method section on page 20, lines 614 – 616: “From literature we could not identify EC degradation pathways for CO₂ production which do not either require impurities such as water or HF^{3,16} nor a direct adsorption of the solvent molecule with the lithium metal surface³². We therefore did not implement any CO₂-producing reaction in our study.”

Reviewer 3:

The authors present a multiscale kinetic Monte Carlo/continuum model to analyze the initial SEI formation when Li metal is in contact with the electrolyte. The activation energies of reactions are calculated from DFT simulations. The layered structure of SEI, the effects of the salt concentration and the EC concentration are studied in this manuscript. The writing of this manuscript conforms to the standard of the scientific paper. The authors walk the reader through assumption technical derivation and some test demonstrations. I praise the authors for focusing on this challenging problem and perhaps this work can be considered as a first step laying out a possible approach. However, some conclusions of the numerical simulations seem fairly trivial. And, the effects of EDL, which are one key point in metal-ion batteries, on the formation of SEI are ignored. Besides, I have some major concerns as follow:

We would like to thank the reviewer for the positive and detailed feedback and valuable suggestions on improving our manuscript. Below, we individually comment in detail on each remark.

Remark 3.1: For the innovation of this method used in this manuscript, the method of KMC + DFT + analytical model is already used in the lithium metal batteries, such as Min Feng et al 2022 J. Electrochem. Soc. 169 090526. The authors should explain the differences and advantages of their method compared with other similar methods. Besides, the authors should add a more comprehensive review of the literature about the similar methods.

We appreciate that the reviewer pointed out this interesting study. We are fully aware that several combinations of kMC and DFT applied to different battery-related questions have been reported before and we reviewed many of these studies in our introduction. We added the mentioned study as another example. However, it should be noted that the mentioned work does not study SEI formation on lithium metal but the effect of pressure on the generation of voids during lithium stripping. Furthermore, to the best of our knowledge, we are the only DFT + kMC/continuum study that investigates the SEI formation on lithium metal up to the reported length- and time scales. In comparison with the mentioned work, we report on a more complex kMC model, which includes 12 directly considered species, 8 reaction processes (partly reversible), 3 clustering processes and transport of all non-solid species. Moreover, we consider a three-dimensional kMC-lattice and directly couple the kMC model with a continuum model in order to ensure global electroneutrality in our system. We are therefore convinced that our modelling approach and results overall provide great novelty compared to previous studies.

Changes made: The mentioned study was added to the introduction on page 3, lines 105 – 107 as follows: “On lithium metal, combinations of DFT and kMC were recently applied to evaluate the Li morphology evolution in porous electrodes [40] and the effect of external pressure on the void generation during stripping [41]”.

Remark 3.2: At present, there are a large number of the experimental results of SEI. The authors should compare the simulation results with the experimental results in this manuscript to show the advantages of their method. Note that the layered structure of SEI, the detailed component of SEI under different electrolytes, and the time evolution of SEI, which can give the readers a more intuitive understanding of the analysis of their method.

There are indeed numerous experimental SEI studies reported. However, only a few investigate the initial SEI formation on lithium metal. Cryo-TEM ([Nano Letters, 20, 1, 418-425 (2020)]) is one of the methods that was recently used to study SEI morphology and composition. It reaches a nm-resolution but was to the best of our knowledge only applied to SEI formation on electrochemically deposited lithium which is another process than the one studied in our manuscript and can therefore not be directly compared. Other techniques such as depth-resolved XPS, TOF-SIMS or FIB-SEM give interesting information on SEI composition and partly morphology but do not reach a nm-resolution which would be needed in order to identify a layered inorganic layer. Moreover, neither of these methods can track the time evolution of SEI formation, especially not with a molecular resolution. Moreover, these experimental methods also struggle to identify the underlying mechanisms of SEI formation. We already briefly discussed this in our introduction.

Changes made: In order to make this clearer and highlight the advantages of our method, we added the following sentences to the results section of our manuscript on page 9, lines 300 – 304:

“Detailed experimental studies on the initial SEI formation on lithium metal and its resulting composition, especially with a sub- μm resolution, are scarce in literature^{4,54}. To the best of our knowledge, there is no experimental study, yet, which was able to reveal the here observed layering of the inorganic SEI. Future advancements in experimental methods may allow to reach a similar resolution and thus provide an experimental validation.”

We further added the following sentence to our ‘Discussion’ section on page 16, lines 491 – 494:

“Thereby, the presented approach is able to track the time-evolution of SEI formation on a molecular resolution and to reveal details of the SEI composition and morphology and on the underlying formation mechanisms, which are presently inaccessible by experiments^{4,56}.”

Remark 3.3: In the part of introduction, the calculation method of the activation energy has some other new methods, such as the hybrid quantum-classical method established in the metal-ion batteries 10.1016/j.jpowsour.2023.232880, and the used machine learning method 10.1063/5.0096027. The authors should cover these works in the introduction.

We agree that combinations of machine learning predicted activation energies and kMC calculations such as in this recent study ACS Energy Lett. 1446–1453 (2022) are a very promising pathway for the future. Therefore, we cited the mentioned machine learning paper as follows in our introduction on page 3, lines 93 – 94:

“Modern machine learning methods could accelerate the prediction of reaction energy profiles for large reaction networks [30,31]”.

Moreover, we now mention that hybrid quantum-classical methods are able to consider the effect of the EDL on intercalation and deintercalation processes on page 3, lines 95 – 98:

“Moreover, different hybrid quantum-classical MD approaches were recently reported that yield SEI compositions and structures for a wide range of electrolytes [33] or allow the prediction of the effect of the electrochemical double layer on intercalation and deintercalation processes [34].”

Changes made: The suggested articles are now cited in our introduction as described in the highlighted sections above.

Remark 3.4: Besides, the effects of EDL on the formation of SEI should be analysed.

This is an interesting point to discuss: We agree that the formation of an EDL could affect the SEI formation in several ways: It has an influence on the ion distribution and hence on the local concentration of salt and Li^+ -ions as well as charged intermediate species. By this, it has an influence on the reaction probabilities and therefore on the dynamics of SEI formation. Vice versa, the EDL constantly changes during build-up of the SEI. At sites of Li that lose direct contact to liquid electrolyte, no mobile species and thus no accumulation of positive ions is expected. Hence, the EDL is expected to vary strongly locally and over time during SEI formation. Once stable SEI layers have formed, we expect double layers at the interface between electrode and solid electrolyte (i.e. SEI) and between SEI and liquid electrolyte. For the case of Li-ion battery anodes, we previously formulated a continuum cell model with a double layer at the electrode/SEI and at the SEI/electrolyte interface, respectively, and used this to determine from experimental EIS the capacitances and the thickness of the SEI (Batteries & Supercaps 5, 7 (2022)). In addition, one has to be aware, that in a battery, typical times for EDL charge/discharge are in the order of milliseconds to seconds, which is far beyond the here considered time scale (J. Electrochem. Soc. 165, 16 (2018)). It is clear, that initial SEI formation, with its strongly changing interfaces causes a complex behaviour and changes of double layer(s) (if one can name it a layer at all).

In our model, the effect of EDL is covered in multiple ways. First of all, since the lithium metal atoms from the anode are oxidized during the simulation, there are many positively charged Li^+ -ions available close to the surface. These are distributed within the simulation box over time by the implemented transport process. Further, the charge of the direct environment is considered in the transport rate calculation (Equation 6) which accounts for local electroneutrality in our simulation box. Last but not least, the double layer capacity is included in our continuum model according to Equation 14. To elucidate what happens at the interface, we plot the charges per layer in the SI and here. During the first 100 ns, the implementation of these effects indeed leads to an accumulation of positive charge at the lithium metal surface. However, after the first passivation this effect vanishes, since less lithium oxidation occurs. An even more elaborate consideration of the EDL would need to consider the constantly changing surface due to Li oxidation as well as all charged intermediate species and would therefore need an extensive model development which is out of the scope of our present study. It would need theory development on various scales, easily filling a further publication. At longer time scales, it might be suitable to apply a continuum approach similar to the one which we developed in our study Batteries & Supercaps 5, 7 (2022) on larger time scales in order to account for the double layer effects at the lithium/SEI and SEI/electrolyte interfaces. Overall, it would certainly be interesting to study this effect in more detail in future investigations.

Changes made: We added the local charge distribution in the kMC box after 0.1 ns, 1 ns, 10 ns, 100 ns and 1 μ s and a brief discussion to the SI on page 7, lines 100 - 121. From this, it can be seen that up to 100 ns there is indeed a local accumulation of positive charges at the lithium metal surface. We further added a reference to the analysis of the EDL in the SI to our main manuscript on page 10, lines 315 – 317 as follows: “Last but not least, the developing electrical double layer could affect the SEI formation. Its effect on our modeling results is analyzed in the SI.”

Figure 1 Charge accumulation during SEI formation as function of height and time, showing the build-up of a double layer at the Li surface, which is subsequently consumed by SEI formation.

Remark 3.5: In actual battery conditions, the surface morphology of the electrode is non-uniform, so the structure of SEI would be affected by the surface morphology. What is the effect of the surface morphology of lithium metal on the formation of SEI in their method? The advantages of this KMC coupled method can be reflected in the non-uniform initial design of the surface morphology.

We agree that on the scale of an actual battery, the surface morphology with surface roughness in the dimension of 0.1 μ m (Adv. Mater. Interfaces, 4, 1700166 (2017)), would most probably have an impact on SEI formation. In our study, we are below this scale: we study a small fraction of the surface of approximately 5 x 5 nm and would not expect such significant differences in the surface morphology on this scale. Moreover, we are not studying SEI formation during plating or stripping in this work, which would lead to major inhomogeneities of the surface morphology.

However, surface morphology changes are induced in our study by lithium metal oxidation reactions and the corresponding electrolyte reduction reactions which lead to SEI formation. Since these anyways lead to ongoing surface reconstructions, we do not expect a significant impact of smaller morphological inhomogeneities in the initial lithium metal surface configuration.

In order to verify this, we performed an additional simulation with the same parameters as applied for the simulation shown in Figure 3, but with a monolayer step in the initial lithium metal crystal structure. The initial configuration as well as the resulting SEI after 1 μ s can be found in Figure S5 in the supporting information. The comparison with the final configuration after 1 μ s in Figure 3 shows that there is – apart from stochastic variations – no significant difference in the formed SEI. The Li_2CO_3 layer still forms below the initial interface with a denser layer of LiF on top. Moreover, the overall thickness of the SEI remains the same. We therefore conclude, that there is no major impact

of smaller surface inhomogeneities on the investigated scale on the formed SEI within the first microsecond after the initial contact of the liquid electrolyte and lithium metal.

Changes made: As described above, we performed a new simulation with a monolayer step in the initial lithium metal crystal structure. We added the initial and final kMC box to the SI in Figure S5 and also provided a short comparison with the SEI formation on a perfectly smooth lithium metal surface on page 8, lines 131 – 140 of the SI. Additionally, we added a short link to the additional simulations in the SI to our main text on page 9, lines 296 – 300 as follows: “The same SEI formation and chemical SEI composition were observed when smaller inhomogeneities were introduced to the lithium metal surface, as shown in Figure S5 in the SI. This shows the low sensitivity of the principle SEI layer on structural inhomogeneities. Since the SEI is so thin, the layer structure can thus be expected also for Li interfaces with surface roughness.”

Remark 3.6: In the part of discussion, the authors have said their method has a 50 times larger time scale and a 32 times larger length scale than comparable ReaxFF MD simulations. I cannot find a clear comparison of the calculated costs between their method and other calculations in this manuscript. The authors should state the calculation cost of their method. Besides, their model should compare not only with the ReaxFF MD simulations, but also with the similar coupled method mentioned in the comment (1).

Thank you for this recommendation, which helps comparing the efficiency of both methods better.

Changes made: In order to give a better estimate of the calculation costs of our method, we added a comparison of simulation time and the respective specification of the used computers for our study and the MD study to our manuscript on page 7, lines 227 – 231:

“Moreover, comparison shows the low computational costs of our approach: According to the authors of the comparative MD study³² their calculations ran for a couple of weeks on high performance computer clusters in order to reach 20 ns. In contrast our kMC/continuum model only took 29.2 minutes on a personal computer with an i7-8700 CPU and 16 GB RAM to reach the same time on a 32 times larger length scale.”

It is more difficult to compare the calculation costs with the cited study since a very different system is investigated and the authors of the previous work do not give detailed information about the calculation costs in their manuscript. Anyways, the calculations in the cited study are on the ps time-scale which is well below the μ s-scale reported in our study.

Reviewer #1 (Remarks to the Author):

The authors have adequately answered the questions and incorporated the revisions in the manuscript.

Reviewer #2 (Remarks to the Author):

Authors have revised the manuscript point by point. I recommend that the present version be accepted for publication.

Reviewer #3 (Remarks to the Author):

In the present study, an in-depth model-based analysis of the initial SEI formation when Li metal is in contact with an ethylene carbonate/ethyl methyl carbonate electrolyte with LiPF₆ as conductive salt, with the purpose of expectation to suggest new design strategies for SEI on lithium metal. I think the method is somewhat new, while there are still some critical issues, and I don't think the manuscript has reached the broad readership and/or provide the significance of SEI in guiding the applications. I could not recommend this manuscript for accepting in Nat Commun. Please see the comments as below:

(1) I think the most of the issue in the present study is lack of experimental validation of the multiscale modelling. For example, some of the output from such model should be verified by a certain experiment of characterizing the SEI films.

(2) I think there is a gap that the authors should bridge. In the present multiscale model, the authors have provided some details of the SEI. However, the authors should discuss the consequence the lithium batteries would be impacted if the SEI is varied.

(3) I believe that the colleague is expected to understand what strategies would be applied to improve the SEI for promoting the lithium-based systems. I would ask the authors to give more details for guiding from physics to chemistry.

Reviewer #1 (Remarks to the Author):

The authors have adequately answered the questions and incorporated the revisions in the manuscript.

Reviewer #2 (Remarks to the Author):

Authors have revised the manuscript point by point. I recommend that the present version be accepted for publication.

We thank the reviewers #1 and #2 for reconsidering our manuscript and for supporting the publication in Nature communications.

Reviewer #3:

In the present study, an in-depth model-based analysis of the initial SEI formation when Li metal is in contact with an ethylene carbonate/ethyl methyl carbonate electrolyte with LiPF₆ as conductive salt, with the purpose of expectation to suggest new design strategies for SEI on lithium metal. I think the method is somewhat new, while there are still some critical issues, and I don't think the manuscript has reached the broad readership and/or provided the significance of SEI in guiding the applications. I could not recommend this manuscript for acceptance in Nat Commun. Please see the comments as below:

We thank the reviewer for taking the time to carefully reconsider our manuscript and would like to address some concerns:

We are confident that the method and results that we report on are of high interest for a broad readership. Once good kinetic data is available the approach can be easily transferred to account for the effect of additives, different electrolyte systems or other anode materials e.g. graphite or hard carbon for Na-based batteries and could therefore be an important computational tool to suggest advantageous electrolyte compositions. In combination with advanced machine-learning algorithms which are able to predict reasonable reaction mechanisms and kinetic data this could even be used as a tool for electrolyte screening. These points have been added to the end of the discussion section.

Moreover, in our study we give unprecedented insights into the molecular mechanisms and limitations that govern the early stages of SEI formation and give suggestions of how to modify the electrolyte composition in order to change the resulting SEI.

Remark 3.1: I think the most of the issue in the present study is lack of experimental validation of the multiscale modelling. For example, some of the output from such model should be verified by a certain experiment of characterizing the SEI films.

We would like to refer the reviewer to our responses to Remark 1.8 and Remark 3.2 in our first revision. The time scales which we report on are a big step forward compared to classical molecular dynamic or density functional studies. Nevertheless, the molecular resolution and also the time scale that we report on are currently inaccessible for experimental measurements, as we also discussed in our 'Discussion' section. A complete experimental validation is therefore at the moment hardly feasible and needs further experimental methods development.

Remark 3.2: I think there is a gap that the authors should bridge. In the present multiscale model, the authors have provided some details of the SEI. However, the authors should discuss the consequence the lithium batteries would be impacted if the SEI is varied.

We thank the reviewer for pointing this out. Our presented model focuses on predicting the chemical composition and structure of the SEI, but it does not predict the performance of the SEI. Based on the reported structures, an additional performance model would be required to reveal the structure-performance relationship. Alternatively, the suggested SEI structures may be experimentally synthesized as suggested by the model, and then tested experimentally to relate the predicted structure to performance. We added this outlook to our discussion section.

Remark 3.3: I believe that the colleague is expected to understand what strategies would be applied to improve the SEI for promoting the lithium-based systems. I would ask the authors to give more details for guiding from physics to chemistry.

In our current study we discuss the effect of different salt and solvent compositions. For this we would like to refer to the section "Influence of salt and solvent concentration" of our manuscript. We show how different electrolyte compositions can affect the SEI formation and its limiting processes and therefore demonstrate how the developed simulation tool can be used for knowledge-driven SEI design. Future studies will e.g. concentrate on the effects of additives, further, more advanced electrolyte systems and artificial passivation layers. We added these points to the end of the discussion section.

We are convinced that this fundamental knowledge is crucial to enable lithium metal anodes in rechargeable batteries with liquid electrolytes.